# Recent Advances in the Investigation of Poly(lactic acid) (PLA) Nanocomposites: Incorporation of Various Nanofillers and their Properties and Applications

**DOI:** 10.3390/polym15051196

**Published:** 2023-02-27

**Authors:** Nikolaos D. Bikiaris, Ioanna Koumentakou, Christina Samiotaki, Despoina Meimaroglou, Despoina Varytimidou, Anastasia Karatza, Zisimos Kalantzis, Magdalini Roussou, Rizos D. Bikiaris, George Z. Papageorgiou

**Affiliations:** 1Laboratory of Polymer Chemistry and Technology, Department of Chemistry, Aristotle University of Thessaloniki, GR-54124 Thessaloniki, Greece; 2Department of Chemistry, University of Ioannina, P.O. Box 1186, GR-45110 Ioannina, Greece

**Keywords:** poly(lactic acid) (PLA), nano-additives, nanocomposites, synthesis, properties, applications

## Abstract

Poly(lactic acid) (PLA) is considered the most promising biobased substitute for fossil-derived polymers due to its compostability, biocompatibility, renewability, and good thermomechanical properties. However, PLA suffers from several shortcomings, such as low heat distortion temperature, thermal resistance, and rate of crystallization, whereas some other specific properties, i.e., flame retardancy, anti-UV, antibacterial or barrier properties, antistatic to conductive electrical characteristics, etc., are required by different end-use sectors. The addition of different nanofillers represents an attractive way to develop and enhance the properties of neat PLA. Numerous nanofillers with different architectures and properties have been investigated, with satisfactory achievements, in the design of PLA nanocomposites. This review paper overviews the current advances in the synthetic routes of PLA nanocomposites, the imparted properties of each nano-additive, as well as the numerous applications of PLA nanocomposites in various industrial fields.

## 1. Introduction

Industrial plastic production was started at the beginning of 1950, and today it is estimated that about 8.3–9.1 million metric tons (Mt) of plastics have been globally produced [1]. Most of these are used in packaging materials (>50%), and are light and very difficult to be recycled; thus, the 79% of produced plastics are discarded in landfills or accumulate in the environment [2]. Most commercially produced plastics are fossil-based materials, are very stable, and degrade slowly in the environment, mainly owing to UV irradiation from the sun, forming fragments of different sizes. With sizes ranging from 1 μm to 5 mm, called microplastics (MPs), they have been recognized in the last 10–20 years as the main threat to living microorganisms [3,4,5] and human health, and can be found in oceans, sediments, rivers, and sewages [6,7]. Fears concerning the effect of MPs in living organisms has lead scientists and companies working with polymers to focus their research activities on the production of eco-friendly plastics that could be degraded in the environment without harmful by-products by substituting them with nondegradable fossil-based plastics. Thus, biodegradable and biobased polymers have garnered huge interest in recent decades as an alternative solution to the growing demand for single-use petroleum-based conventional polymers [8].

The ever-growing global concern regarding the environmental impacts and raising awareness towards issues such as global climate change and plastic waste management has led consumers, and in turn fuelled industries and governments, to adopt a more sustainable approach [9]. Many waste management solutions, such as reusing, reducing, composting, and recycling, have contributed to a decrease in white pollution (i.e., single-use plastic contaminating the environment, reaching nature, oceans, and landfills), and the development of biodegradable polymers has been a favourable change, since they can be recovered through recycling and composting. In this regard, PLA has been the frontrunner in (bio)polymers since it presents a potential solution to the waste disposal problem [10,11]. Lactic acid (LA) is the precursor to PLA and can be obtained by the fermentation of renewable resources such as corn, cassava starch, and potato [12].

PLA, a linear aliphatic polyester, exhibits a lot of attractive properties and thus is facilitated in numerous commercial sectors [13,14]. Its excellent barrier to flavor and good heat sealability have increased its usage in the food packaging field, while its appealing mechanical properties, high clarity, and ease of processing have expanded its utility in medical, automotive, textile, and agriculture areas [15,16]. In addition, it is renewable, compostable, and carries a relatively low cost [11]. PLA can be also degraded slowly in the environment since is not a biodegradable polyester but a compostable, leading to the production of microplastics. From several toxicity studies it was found that PLA MPs did not show any significant effects on plants and soils [17], while in the case of the human body, PLA is a safe biopolymer, since it is used extensively in medical and biomedical applications [18]. It can be slowly degraded in the human body, producing non-toxic fragments such as lactic acid [19].

In spite of these advantages, PLA also presents some drawbacks such as fragility, high permeability to gas and vapor, low melt strength, and thermal stability, while its degradation rate is slower compared to other common natural organic wastes, such as food and yard waste. This results in a significant limitation in its acceptance into industrial food and yard composting facilities and generally in its widespread use [20]. Due to the significant improvement observed in hybrid PLA nanocomposites, even at very low nanofiller loadings, many investigations have been carried out concerning this matter [21,22]. Hybrid nanotechnology has created a new revolution in the area of material science, developing the most high-tech advanced composites for future applications. At least one particle in the nanoscale dimension is included in the nanocomposites. The addition of nanoparticles (NPs) demonstrates remarkable enhancement in thermal, mechanical, physical thermo-mechanical properties due to better distribution, a highly specific ratio, and effective polymer–filler interaction [8].

Various nanoparticles have been used for the development of PLA-based nanocomposites. Among them, clays, carbon, bioglasses, metals, and natural nanomaterials are the most common [23,24,25]. This paper provides a comprehensive overview of the state-of-the-art PLA nanocomposites, including past and recent achievements that could influence future research and industrial applications. It presents information about the synthesis of PLA nanocomposites and the effect of nanomaterials on the morphology and physicochemical properties (mechanical, thermal, biological, electrical, biodegradable properties, etc.) of the final materials [26].

## 2. Synthesis of Nanocomposites

PLA nanocomposites are generally produced by means of three main techniques, namely in situ polymerization, solution casting, and melt mixing processing, and to a lower extent by electrospinning and additive manufacturing. Figure 1 overviews the different preparation methods for the incorporation of various nano-additives into the PLA matrix.

### 2.1. In-Situ Polymerization

In-situ polymerization involves the dispersion of nanoparticles in a liquid monomer or monomer solution followed by polymerization, accomplished with the help of heat, radiation, or suitable initiators [27]. This process can be used to synthesis polymer nanocomposites under controlled conditions in order to produce materials with known and homogeneous compositions as well as with covalently bond formation between nanoparticles and the polymer matrix [28,29]. It has been proven by several studies that the in-situ polymerization assists in raising the interparticle spacing, achieving uniform dispersion of nano-reinforcements in the selected polymer matrix. Moreover, it controls the nanoparticle exfoliation, even at higher concentrations with a lesser chance of agglomeration [11]. Additionally, during the synthesis of PLA nanocomposites via in situ polymerization, covalent bonds between the growing polymer chains and the active groups of nanoparticles are formed, enhancing the PLA properties [15,27]. Manju et al. [30] reinforced PLA with sepiolite nanoclay via in-situ polymerization and obtained agglomerate-freed bio-nanocomposite even at higher loading of the silicate (7 wt.%). The melting point of the PLA was increased on incorporation of the clay due to homogenous dispersion attained by the excellent interaction of C-O of PLA with the clay’s Si-OH groups [15]. Yang et al. [31] synthesized poly(L-lactide) (PLLA)-thermally reduced graphene oxide (TRG) via the in-situ ring-opening polymerization of lactide, with TRG as the initiator. They proved that the chemical interaction between PLLA and the TRG sheets was strong enough to increase the nucleation rate and the overall crystallization rate of the PLLA. Song et al. [32] grafted PLA covalently with the convex surfaces and tips of the multi-walled carbon nanotubes with carboxylic functional groups (MWCNTs–COOH) via in situ polycondensation. PLA was grafted on MWCNTs–COOH, which acted as an initiator, and the TGA results showed that the grafted PLA content could be controlled by the reaction time.

### 2.2. Solution Casting

In this technique, a solution of the polymer in a suitable solvent is mixed with a dispersion of the nanoparticles in the same or a different solvent. In the production of PLA nanocomposites, the solvent is usually used to pre-swell the nanofillers and the dispersion of the swollen nanofillers is mixed with the PLA solution. The sonication method is often practiced, aiding in the dispersion of nanofillers [33]. When the solvent is evaporated, the adsorbed polymer on the swelled nanofillers will reassemble, sandwiching the polymer to form nanocomposites. The solution method is an effective fabrication technique for nanocomposites due to the ease of processing nanoparticles in solvent, and it can be mainly used for nanoparticles that are thermally sensitive and degrade at high temperatures. On the other hand, in a practical sense, the solution intercalation approach is unattractive as a result of the use of organic solvents, and the polymer may be less serviceable as a result of the traces of solvent that could contaminate it [9,15,27]. The solution casting method is the conventional method for manufacturing PLA/TiO_2_ nanocomposites. Feng et al. [34] fabricated PLA/TiO_2_ nanocomposites, using different organic solvents and dissolution conditions. Specifically, PLA was dissolved in a chloroform solution, and magnetically stirred at 40 °C until complete. Then, 2 wt.% TiO_2_ were dissolved in absolute ethanol to obtain five different nanocomposites. After ultrasonic dispersion for 60 min, the mixture was slowly added dropwise to the stirred PLA solution. The results showed that the maximum tensile strengths of the nanocomposites were achieved. Moreover, the incorporation of TiO_2_ significantly decreased the water vapor transmittance rate of the nanocomposites while improving the water solubility. Fu et al. [35] employed citric acid-modified nanoclay to enhance PLA through solution casting to fabricate a shape memory nanocomposite. They showed a considerable restriction in the PLA chain segment, which supported an anisotropic force required in shape recovery. The grafted citric acid on the clay and ester groups of the PLA provided good affinity, being responsible for the shape memory property.

### 2.3. Melt Mixing

Melt mixing is a technique wherein a high temperature, higher than the melting point of the PLA, under the shear force is applied to the mixture of PLA matrix with nanofillers to prepare nanocomposites with the objective of obtaining well-dispersed nano reinforcement. However, it requires a pre-drying process of PLA granules as PLA has a high sensitivity to moisture [36].

This process presents significant advantages over solution casting and in situ intercalative polymerization methods. It supports a high volume of bulk polymer, requires no solvent/chemical, and carries a relatively low cost, faster than the solution and in situ polymerization and with the least environmental impact. Moreover, this method is compatible with processing techniques such as injection molding and extrusion, which is used in the polymer industry. Nevertheless, melt mixing subjects PLA to some level of degradation due to applying elevated temperature and mechanical shear during the process [37]. Melt blending has been widely used to prepare PLA/nanoclay composites. Ray et al. [38] prepared PLA nanocomposites containing 3, 5, and 7 wt.% OMMT (MMT modified with octadecyl ammonium cation), in a twin screw extruder (TSE), which was then converted into 0.7–2 mm thick sheets for characterization studies. On the other hand, this method is unsuitable for PLA/graphene nanocomposites since the pristine graphene has a tendency to agglomerate in the polymer matrix, unlike the other preparation routes of PLA/graphene nanomaterials that include in situ intercalative polymerization and solution intercalation [39]. Stacking occurs due to the large ratio of surfaces of graphene sheets in relation to their thickness, which yields significant van der Waals forces and a strong interaction between single sheets of graphene. The physical and chemical properties of such graphene aggregates are similar to the properties of graphite with relatively small surface areas [40]. The preparation of polymer-based nanocomposites has become a great challenge in obtaining a good distribution of the nano-reinforcement. Nonetheless, as for graphene or graphene oxide, the formation of bundles is not a problem, although the inclination of incomplete exfoliation and restacking can occur. The graphene functionalized through modification is preferred for melt intercalation [27]. This has been supported by Murariu et al. [41], who produced PLA with commercially available expanded graphite. The expanded graphite nanofiller yielded PLA composites with competitive functional properties such as high rigidity, with the Young’s modulus and storage modulus increasing with the nano-additives’ content.

### 2.4. Electrospinning

Recently, PLA fibers have been developed to be applied in the textile industry, for medical applications such as tissue engineering, and in food packaging applications. PLA nanofibers fabricated with the electrospinning method present a higher surface area and porosity than regular fibers and are applied to covering filtration systems, sensor assembly, tissue scaffolds, and protective clothing. In this method, the choice of solvent and the concentration of polymer and incorporation of nanofillers in the process affect the melt or solution spinning and thus the morphologies of the fabricated fibers and pores [42]. Melt spinning and solution spinning are traditional processes of PLA fiber fabrication but are mostly limited to fiber diameter on a micro-scale, which is a constraint to further PLA applications. Another drawback of these processes is that some active components such as anti-bacterial and antioxidant ingredients are thermolabile, which makes conventional melt processing unsuitable for preparing such nanocomposites [43]. On the other hand, electrospinning is a promising and versatile technique for producing multifunctional nanosized PLA nanocomposite fibers through the application of high voltage to viscoelastic polymer solution/melt with varying nano-reinforcements. The electrospinning machine set-up can be a single-nozzle or multi-nozzle type [44,45]. This technique can produce fibers with diameters ranging from the nanoscale to the submicron scale, which are typically one to two orders of magnitude lower than those of traditional solution-spun fibers. Specifically, Touny et al. [46] fabricated PLA/HNT nanocomposite fibers via electrospinning and the average diameter of the fibers spun was 230–280 nm. Shi et al. [47] produced PLA/cellulose nanocrystals (CNC) electrospun fibers and they found that PLA nanocomposite fibers with 1 wt.% CNC-loading was 1 μm while they decreased to 642 and 405 nm at 5 and 10 wt.% CNC-loading levels, respectively. In another work, Liu et al. [48] fabricated electrospun PLA/GO and PLA/nHAp fibers and they proved that the addition of 15 wt.% nHA in the PLA matrix produced fibers with a 563 ± 196 nm diameter size, while the incorporation of 3 wt.% GO decreased the diameter of the composite fibers to 412 ± 240 nm. The PLA nanofibers fabricated with the electrospinning method also present a higher surface area and porosity than regular fibers and are applied to covering filtration systems, sensor assembly, tissue scaffolds, and protective clothing. In this method, the choice of solvent and the concentration of polymer and incorporation of nanofillers in the process affect the melt or solution spinning, and thus the morphologies of the fabricated fibers and pores. PLA nanocomposites have also been used to fabricate nanofibers via electrospinning; however, some researchers have experienced difficulty in the homogeneous generation of nanocomposite fibers due to the difficulty of nano-additives’ dispersion in the PLA matrix as a result PLA nanocomposites not being able to be electrospun.

Kim et al. [49] used an extremely fine hydroxyapatite powder, which was prepared using the sol–gel process, to improve the dispersion of the ceramic powder within the PLA matrix during electrospinning. Zhang et al. [50] ultrasonically dispersed ZnO nanoparticles in a PLA matrix and developed electrospun membranes with enhanced antimicrobial activity and UV blocking for food packaging applications. Xiang et al. [51] fabricated PLA/cellulose nanocomposite fibers electrospun at elevated temperatures from solutions of PLA in N,N-dimethylformamide (DMF) containing suspended cellulose nanocrystals. They optimized the electrospinning conditions to form PLA/cellulose nanocomposite fibers with uniform diameters; however, they found that the average fiber diameter decreased with the addition of cellulose nanocrystals. They also confirmed the presence of cellulose nanocrystals at the surface of PLA by surface elemental composition analysis.

### 2.5. Additive Manufacturing

Additive manufacturing (AM) is a very promising and rapid processing method permitting the manufacture of 3D-printed models or functional parts with complex geometries [52]. Fused deposition modeling (FDM) is a widely adopted AM technique due to its simplicity and low material wastage and cost. This process begins by electronically designing and “slicing” the 3D model of the item in the CAD to be loaded into the FlashPrint software, which generates the printing path. FDM is an extrusion-based process wherein material is selectively dispensed via a shaped needle or orifice [53]. The nozzle temperature has to be set above the melting temperature of the thermoplastic for it to flow. In a recent optimization study, it was found that the best properties of pure PLA prints were achieved when the printing temperature was 215 °C. Additionally, the content of water in the PLA composite filament should be less than 50–250 ppm, otherwise the presence of larger amounts of water can cause swelling of the filament, which can block the hot nozzle during the 3D process. However, these characteristics of PLA are affected by the increasing of different amounts and types of nanofillers. Moreover, the morphology of the PLA nanocomposites and the interactions between polymer—polymer and polymer—nanoparticles affect the final properties of the 3D-printed product [54].

Wang et al. [55] researched the capacity of PLA with nHAp (at a content of 0%, 10%, 20%, 30%, 40%, and 50%) to be 3D printed for medical applications. The results showed that the PLA/nHAp composite had a stable structure, and the processing of this material was highly controllable. When the proportion of nHAp was greater than 50%, the composite material could not pass through the printing nozzle coherently and stably due to its high brittleness. When the nHAp ratio was less than or equal to 50%, the composite filament material could be printed by FDM. After applying the 3D printing process of PLA/nHAp composites, the printed samples with 0% to 30% nHAp could still maintain the integrity of the structure after the pressure test, which proved that these samples maintained their elasticity. However, when the n-HA ratio reached more than 40%, the printed sample became brittle and could not maintain complete shape after compression. In a recent study by Yang et al. [56], a filament based on PLA/CNTs composites was produced by the FDM process. The effects of the CNT content on the crystallization-melting behavior and melt flow rate were tested to investigate the printability of the PLA/CNT. The results demonstrate that the CNT content has a significant influence on the mechanical properties and conductivity properties. The addition of 6 wt.% CNT resulted in a 64.12% increase in tensile strength and a 29.29% increase in flexural strength. The electrical resistivity varied from approximately 1 × 10^12^ Ω/sq to 1 × 10^2^ Ω/sq for CNT contents ranging from 0 wt.% to 8 wt.%. Three-dimensional PLA-G electrode properties allowed for graphene surface alteration and electrochemical enhancement in the sensing of molecular targets. Bustillos et al. [57] fabricated hierarchical nanocomposites fused by deposition modeling printing using filaments of PLA and PLA-graphene (Figure 2). Enhanced creep resistance in the PLA-graphene was attributed to the restriction of the polymeric chains by graphene, caused by low strain rates identified during secondary creep. Three-dimensional-printed PLA-graphene exhibited a 20.5% improvement in microscale creep resistance as compared to PLA at loads of 90 mN. Wear resistance was increased by a 14% in PLA-graphene as compared to PLA.

## 3. Effect of Nano-Additives on PLA/Nanocomposite Properties

### 3.1. PLA/Metal Oxides

Interest in the investigation of PLA/metal oxides containing nanoparticles has rapidly grown in recent years because of their numerous properties. Particularly, the design of metal-containing nanocomposites is a current trend in scientific research due to their high-potential practical application of antimicrobial and antiviral materials [58]. A series of metal oxide nanoparticles have been synthesized including silica dioxide (SiO_2_), titania dioxide (TiO_2_), zinc oxide (ZnO), magnesium oxide (MgO), boron oxide (BO), and aluminum oxide (Al_2_O_3_). They exhibit a variety of morphologies such as sphere, triangular, star, nanotubes, nanorods, nanowires, etc. The preparation of polymer-metal oxide nanocomposites with a nanophase-separated structure requires the homogeneous dispersion of the metal oxide NPs as well as a reduction in the size of the polymer–metal oxide interface, since it may alter the physical property of the nanocomposites [59]. Due to a high density and limited size, metal oxide NPs present interesting results in terms of chemical and physical properties. In general, properties including thermal stability, toughness, glass transition temperature, optical, tensile strength, etc. are found to be improved by the formation of such nanocomposites with PLA. Therefore, these nanocomposites are widely used in numerous applications, from active surface coating to biomedical applications including structural materials [13,60]. This chapter provides a general overview of the recent studies involved in the fabrication of PLA/metal oxide nanocomposites.

#### 3.1.1. Green Synthesis of Nanosized Metal-Oxides

Over the last decade, novel preparation routes/methods for nanomaterials (such as metal nanoparticles, quantum dots (QDs), and nanofibers and their composites) have been an interesting field in nanoscience [61]. To obtain nanomaterials of desirable shape, size, and properties, two different fundamental principles of synthesis are required. Figure 3 illustrates the top-down and bottom-up preparation routes investigated in the existing literature.

Nanoparticles are prepared through size reduction, disintegrating from the bulk material into fine particles in the top-down approach. This route can be accomplished through physical and chemical methods via lithographic, mechanical milling, sputtering, chemical etching, pulsed laser ablation, and photoreduction techniques. Still, the major downside in the top-down approach is the imperfection of the surface morphology. In the bottom-up approach, also known as the self-assembly approach, the nanoparticle synthesis relies on wet chemical methods (e.g., chemical reduction/oxidation of metal ions) and others, such as sol-gel chemistry, chemical vapor deposition (CVD) co-precipitation, microemulsion, laser and spray pyrolysis, hydrothermal, aerosol, and electrodeposition methods. In the bottom-up synthesis, the nanoparticles are built from smaller units, for example, by atomic and molecular condensation. However, they still suffer from the involvement of potentially hazardous and toxic substances, high-cost investments, environmental toxicity, high energy requirements, lengthy reaction times, and non-eco-friendly by-products.

Interestingly, the morphological parameters of NPs may be moderated by varying the concentrations of chemicals and reaction conditions (e.g., temperature and pH). Nonetheless, if these synthesized nanomaterials are subject to the actual/specific applications, then they can suffer from the following challenges: (i) stability in hostile environments, (ii) bioaccumulation/toxicity features, (iii) expansive analysis requirements, and (iv) recycle/reuse/regeneration. Nanoparticles prepared using heavy radiations and toxic substances such as reducing and stabilizing agents are toxic in nature, thus leading to harmful effects on humans and the environment.

The majority of chemical methods often exploits hazardous reducing agents such as sodium borohydride (NaBH_4_), ethylene glycol, hydrazine, and sodium citrate, but they result in poor control over particle size and size distribution; thus, extra-stabilizing agents such as ascorbic acid, polymers, ligands, dendrimers, surfactants, phospholipids, and surfactants are necessary to avoid the agglomeration of nanoparticles. A high thermal decomposition treatment then takes place to eliminate the reducing and stabilizing agents adsorbed on the nanoparticle surface, which may block the active sites of particles, consequently resulting in a decreased surface area and deteriorated reactivity of the products. Finally, this procedure produces a low yield of the targeted products.

Green nanotechnology has developed recently with various processes to restrain environmental damage by replacing the usage of harmful compounds. More and more interest has been gained by green extraction of metal oxide NPs in the last decade, owing to the use of eco-friendly reagents such as microorganisms, fungi, enzymes, leaves, plants, etc. NPs are synthesized by this eco-friendly synthesis method, which involves a single-step pollution free method that requires less energy to initiate the reaction and the preparation time is less when compared to other synthetic routes. The utilization of ideal solvent systems and natural supplies (such as organic systems) is vital to achieve this aim.

Green synthesis methodologies based on biological precursors depend on various reaction parameters such as temperature, pressure, solvent/solvents system, and pH environment (basic, acidic, or neutral). Biosynthesis is a green synthetic approach that can be categorized as a bottom-up approach where the metal atoms are arranged into clusters and eventually into NPs. Amongst the accessible green methods of synthesis for metal/metal oxide NPs, the exploitation of plant extracts is a rather simple and easy process to produce nanoparticles on a large scale relative to bacteria- and/or fungi-mediated synthesis (Figure 4). Moreover, the plentifulness of effective phytochemicals in various plant extracts, especially in leaves, such as ketones, aldehydes, flavones, amides, terpenoids, carboxylic acids, phenols, and ascorbic acids, is the main reason for their wide extraction. These products are most commonly known as biogenic nanoparticles and have the capability of reducing metal salts into metal NPs.

Ramanujam et al. [63] extracted MgO NPs using Emblica officinalis fruit extract. The mean size of MgO NPs was found to be 27 nm. Ashwini and his group [64] conducted a study upon extracting MgO NPs using aloe vera. The average size of obtained MgO nanoparticles was found to be 50 nm.

**Figure 4 polymers-15-01196-f004:**
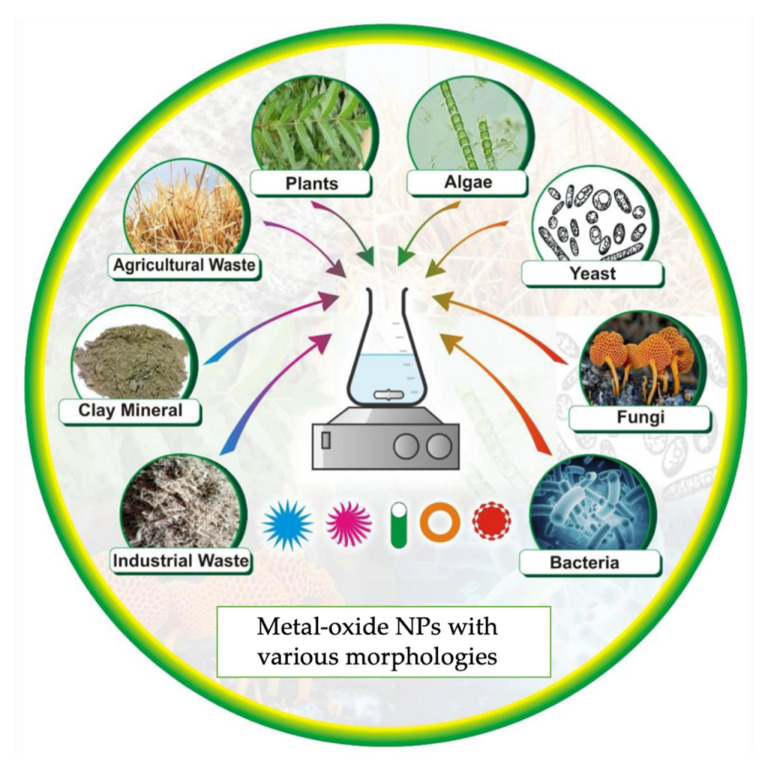
Biological resources for the green synthesis of metal-oxide NPs [65].

Sankar et al. [66] effectively synthesized TiO_2_ nanoparticles e from the aqueous leaf extract of Azadirachta indica with an average size of 124 nm. TiO_2_ NPs using extracts of Echinacea purpurea Herba were biosynthesized using an aquatic solution of Echinacea purpurea herba extract as a bioreductant by Dobrucka [67]. The size of the TiO_2_ nanoparticles was found to be in the range of 120 nm with a spherical size and the presence of agglomerates. The eco-friendly synthesis of TiO_2_ NPs using different biological sources are summarized in Table 1.

Saraswathi et al. [72] biosynthesized ZnO NPs mediated by the leaves of Lagerstroemia speciosa with an average particle size of 40 nm (Figure 5). Interestingly, the obtained NPs had a spherical shape at 200 °C and were rod-shaped at 800 °C (Figure 6).

Yuvakkumar et al. [73] reported for the first time the sustainable biosynthesis of ZnO nanocrystals employing *Nephelium lappaceum* L. peel extract as a natural ligation agent. Green synthesis of zinc oxide nanocrystals was carried out via zinc-ellagate complex formation using rambutan peel wastes. The green syntheses of ZnO nanoparticles using various biological supplies are summarized in Table 2.

Rahimzadeh et al. [79] described a green synthetic process to obtain SiO_2_ NPs utilizing *Rhus coriaria* L. extract and sodium metasilicate (Na_2_SiO_3_·5H_2_O) under reflux conditions. The FESEM images revealed that SiO_2_ NPs are spherical in shape with a minimum degree of agglomeration, and the NP sizes were between 10~15 nm. Rezaeian and his group prepared ZnO NPs from olive leaves with an average particle size between 30–40 nm and spherical in shape [80]. Rahimzadeh et al. [79] in their work synthesized silica NPs via green synthesis by utilizing leaf biomasses from sugarcane (Saccharum ravannae), weed plants (Saccharum officinarum), and rice plants (Oryza sativa) via a simple chemical method. NPs from sugarcane were hexagonal in shape with a 29.13 mean size, particles extracted from weed plants were spherical in shape with average size of 39.47, and nanoparticles from rice plants had a spherical shape with an average size of 30.56 nm.

#### 3.1.2. MgO

MgO is a widely available metal-based oxide used for the reinforcement of PLA. It is a non-toxic alkaline earth metal, reproducible on a large scale, is environmentally friendly, low-cost, and is available in the form of natural periclase. Presently, MgO is listed as a generally recognized as safe (GRAS) material by the US Food and Drug Administration (US-FDA) [81]. Medically, MgO is used for antibacterial purposes, bone tissue regeneration, as well as cancer treatment [82]. Leung et al. [83] reported strong antibacterial activity of the MgO NPs even during the absence of any ROS production. Moreover, the chemical properties and the high surface area of nanosized MgO expanded its usage in numerous modern fields. In brief, MgO is an insulator exhibiting a band gap of E_g_ = 7.8 eV and a high dielectric constant. The MgO NPs that are fabricated using sol-gel and flame spray pyrolysis techniques presented high efficiencies in textiles and methylene blue dye removal. In addition, very fine MgO NPs of E_g_ = 4.45 eV were prepared via a combustion route using urea-formaldehyde as fuel [84]. The transition from insulators to semiconductors is due to the quantum size effects and surface effects in MgO NPs. These effects lead to discrete energy levels in the band gap, which result in its reduction [85].

#### 3.1.3. ZnO

ZnO is a naturally occurring oxidic chemical found in the rare mineral zincite that crystallizes in the hexagonal wurtzite structure P63mc. Metallic zinc is plentiful in the earth’s crust and creates pure powder. Before the invention of nanotechnology, ZnO was used in bulk, but later it was also facilitated as a nanosized material for its intended properties [86]. Nanometric ZnO can appear in various structures. A dimensional structure consists most extensively of needles, hexagons, nano-rods, ribbons, belts, wires, and brushes. ZnO can be produced in two-dimensional structures, such as nanoparticles, nanosheets, and nanoplates. The variety of nanostructures given for ZnO have created more possibilities in the field of nanotechnology [87].

Amongst various semiconducting nano-scale materials, ZnO is a distinctive electronic and photonic wurtzite n-type semiconductor with a wide direct band gap of 3.37 eV and a high exciton binding energy (60 mV) at room temperature [88]. The high exciton binding energy of ZnO allows for excitonic transitions even at room temperature, which could mean high radiative recombination efficiency for spontaneous emission as well as a lower threshold voltage for laser emission. The lack of a centre of symmetry in wurtzite, combined with large electromechanical coupling, results in strong piezoelectric and pyroelectric properties, and thus the use of ZnO in mechanical actuators and piezoelectric sensors [89].

Furthermore, ZnO is biosafe and biocompatible and hence may be employed in biomedical applications without being coated. Statistics showed that approximately 99,000 scientific papers in the period of 2018–2020 were related to the search for new antimicrobial compounds. About 6% of these papers addressed ZnO-related compounds. ZnO is one of the most promising nanomaterials, owing to its excellent antimicrobial properties, good chemical stability, biocompatibility, optical and UV shielding abilities, low price, and widespread availability. ZnO NPs exhibit antimicrobial capability against a wide range of microorganisms such as *E. coli*, *Pseudomonas aeruginosa*, and *S. aureus*, becoming appealing for various applications such as packaging, textiles, and medicine [90].

#### 3.1.4. TiO_2_

TiO_2_ or titania is a renowned and extensively studied material due to its excellent photostability, photocatalytic activity, non-toxicity, safety, chemical stability, outstanding antimicrobial properties, and significant antibacterial properties [91]. Three distinct phases or “crystalline polymorphs” of TiO_2_ NPs are found in nature, namely anatase, rutile, and brookite. Amongst these three, rutile is the steadiest phase, while in contrast, brookite and anatase are the labile phases at all temperatures. The anatase phase of TiO_2_ NPs is the most responsive one, in a chemical manner. Both anatase and brookite exhibit the capability to convert into a rutile phase when heated [92]. Regarding the manufacturing methods of PLA/TiO_2_ nanocomposites, in the literature, a great variety of methods and techniques is reported. The conventional one is the “solution casting method”, in which the nanocomposites are produced by constant dispersion of the NPs in a polymer matrix. Also frequently employed is the electrospinning technique, which is important for the surface coating of in situ-produced NPs, particularly on the exterior or in the majority of a polymer matrix [34] and the “solvent casting method”, followed by a or some hot-pressing steps [93]. The “Sol-gel method” serves as a doable method to increase the dispersibility of TiO_2_ during the film-forming process and to additionally alter the thermal/mechanical properties and antimicrobial action of biopolymer films. Improving the distribution degree of metal oxides in a polymer matrix essentially affects the photocatalytic ability as well as the antibacterial efficacy and mechanical properties [91]. The “Thermally induced phase separation (TIPS) method” [94], “a spin coating process” [95], melt blending through a corotating twin-screw extruder [96], and the “breath figure (BF) method” [97] are also mentioned as potential production methods.

TiO_2_ is the most extensively investigated photocatalyst due to its strong oxidizing abilities, super hydrophilicity, long durability, chemical stability, low cost, nontoxicity, and transparency under visible light [98]. In the last 30 years, the photocatalytic efficiency of nanosized TiO_2_ at degrading organic contaminants in the air and water has resulted in their utilization in several environmental applications. These include photocatalytic hydrogen production via water splitting, degradation of organic pollutants in wastewater photocatalytic self-cleaning, bacterial disinfection photo-induced super hydrophilicity, as well as in photovoltaics and photosynthesis. Currently, water contamination by organic compounds, viruses, bacteria, and metals has been highlighted as a major worldwide problem. In terms of antimicrobial activity, TiO_2_ NPs appeared to be able to eliminate a broad range of micro-organisms, such as fungi, protozoa, Gram-positive and Gram-negative bacteria, viruses, and bacteriophages. Numerous techniques have been explored to remove these pollutants from water, including adsorption and photocatalysis, which comprise eco-friendly routes [99]. Studies have shown that one of the best ways to utilize photocatalytic TiO_2_ NPs is to immobilize them onto the surface of polymeric materials.

This excellent photocatalytic ability of TiO_2_ has also enabled its use in food safety applications, such as packaging materials, cutting boards, conveyor belts, or non-contact surfaces such as floor, walls, and curtains. In the presence of a UV-A light source, titania has the capability to produce the transition of an electron towards the conductive band, supporting the oxidative capacity of other species by generating reactive oxygen species (ROS) such as superoxide radicals (O·^−2^) and hydroxyl radicals (·OH) [98,100].

#### 3.1.5. n-SiO_2_

Advancement in nanotechnology has led to the production of a nanosized silica, SiO_2_, which has been widely used as filler in engineering composites. It is often added as a cementitious material since it can reduce the use of cement due to its reinforcing effect, as well as accelerate the hydration and alter the microstructure evolution of cementitious composition [101]. Many scientific works have been conducted regarding the incorporation of n-SiO_2_ in the PLA matrix due to its numerous properties. Moreover, nano SiO_2_ provides environmental protection, safe and non-toxic functions, abundant surface groups, a huge specific area, and a high rigidity and modulus, leading to its wide usage as a cheap but effective nanofiller for polymers. Moreover, the nontoxic nature, the high thermal and chemical stability, water resistance, and mechanical properties as well as the important role in the biomineralization of SiO_2_ render it a biocompatible nanofiller that retains the biocompatibility of the PLA matrix. In addition, nano SiO_2_ has a white color, so the composite filament is bright and can be colored with almost any desirable color. Statistics show that approximately 99,000 scientific papers in the period from 2018 to 2020 were related to the search for new antimicrobial compounds. About 6% of these papers addressed metal-related compounds [90].

### 3.2. Carbon-Based PLA/Nanocomposites

#### 3.2.1. Carbon Nanotubes (CNTs)

In spite of PLA’s advantages, its mechanical as well as electrical and thermal properties could be further improved so as to expand the application fields. The most effective way to overtake this disadvantage is the filling of a PLA matrix with the introduction of nanoscopic dimensions of carbon-based fillers (Figure 7) with a high aspect ratio, such as carbon nanotubes (CNTs), carbon nanofibers (CNFs), graphene nanoplatelets, and spherical nanoparticles. Upon good dispersion in the PLA matrix, such nano-additives have been found to generate an increase in both the degree and the rate of crystallization, since they provide additional sites for crystallization; in other words, they serve as additional crystallization nuclei [102,103,104]. Their excellent mechanical properties, such as high modulus in the direction of the nanotube’s axis and excellent electrical conductivity that varies from insulating to metallic and the hollow structures of CNTs, have expanded their usage as fillers [105]. The growing interest towards carbon-based nanomaterials, including CNTs, mainly relates to their intriguing properties as conductive fillers for the fabrication of electric/electronic devices, for which production volumes have become dramatically elevated in recent years [106]. CNTs exhibit a highly specific surface area that allows for low loadings to tune the polymer key properties concerning their mechanical, thermal, electrical, and biological performance. CNTs that possess a wall structure consisting of a single graphite sheet closed in a tubular shape are called single-walled carbon nanotubes (SWCNTs), while those consisting of a plurality of graphite sheets each arranged into a tubular shape and nested one within the other are named multi-walled carbon nanotubes (MWCNTs). MWCNTs consist of smaller diameter single-walled tubes inside larger diameter tubes and may vary from a double-walled nanotube to as many as fifty concentric tubes, exhibiting diameters varying between 2 and 100 nm [107]. They have exceptional mechanical properties, aspect ratio, electrical and thermal conductivities, and chemical stability, and hence are considered excellent candidates for the creation of multifunctional materials [108,109].

#### 3.2.2. Graphene

In the case of graphene, it is naturally derived from graphite, which is composed of a layer of less than 100 nm and can be divided into sheets (Figure 7c) around 1–2 nm thick. The physical structure of graphene with its large specific surface area makes it have a better reinforcing effect than other nanofillers [113,114]. Graphene has high mechanical strength and electron mobility, and thus may be facilitated as a filler, even at low amounts in a polymer matrix due to its highly specific surface area and chemical interaction with the matrix by the formation of strong bonds [115,116]. Due to its two-dimensional arrangement of sp^2^-bonded carbon atoms, graphene has been shown to enhance the wear resistance and reduction of friction. Graphene’s incorporation of polymer matrices has resulted in composites exhibiting superior mechanical strength while retaining their flexibility, as well as tailorable thermal and electrical conductivity as a consequence of the generated graphene network in the matrix [57]. Another form of graphene that can be used as nanofillers are graphite nanoplatelets (GNPs) or graphite nanosheets (GNS), which also possess great mechanical, thermal, and electrical properties [117]. The aforementioned fillers of graphene or its derivatives (graphene oxide, reduced graphene oxide) are utilized in varying concentrations as fillers in a PLA matrix in order to prepare nanocomposites with enhanced properties via different techniques [116,118]. Based on the unique properties of graphene, graphene-based polymer composites are expected to offer enhanced electrical and thermal conductivity, improved dimensional stability, higher resistance to microcracking, and increased barrier properties above the matrix polymer [108].

#### 3.2.3. Carbon Nanofibers (CNFs)

Nanosized carbon-based reinforcements such as carbon nanofibers (CNF) into pure thermoplastic matrices have proven to be valuable for manufacturing polymer matrix composites with enhanced mechanical performance and functionality. Carbon nanofibers have a highly specific area, elasticity, and great strength due to their nano-sized diameter. Concerning CNFs, they are characterized by diameters ranging between 50 and 200 nm, being different from the conventional carbon fibers that have diameters in the order of micrometers. Due to the preparation process employed, CNFs possess improved properties, while they can offer property enhancements similar to CNTs in a more cost-effective way [103]. Therefore, they are utilized as fillers in order to enhance mechanical and electrical properties as well as the thermal conductivity of polymers [119]. Critical parameters for enhancing the mechanical and electrical properties are weight fraction of the filler, filler length, and its orientation. Overall, carbon-based nanomaterials offer the possibility to combine PLA properties with several of their unique features [108].

#### 3.2.4. Fullerene

Fullerene is a unique nano-allotropic form of carbon. Similar to carbon nanoparticles (graphene, carbon nanotubes, nanodiamonds), fullerene has been reinforced in polymeric matrices [120,121,122,123]. Fullerene and derived nanofillers affect the structural, electrical, thermal, mechanical, and physical properties of polymeric matrices. However, the main encounter in the formation of the polymer/fullerene nanocomposite is the dispersion and miscibility with the polymeric matrices. Studies have shown that the incorporation of carbon-based nanofillers with various dimensionalities such as zero-dimensional (0D) fullerenes offers an effective approach to PLA nanocomposites with synergistic enhancements in the electrical and mechanical properties when exposed to external stimuli [124]. For that reason, polymer/fullerene nanocomposites have been applied in olar cell, super-capacitor, electronic, and biomedical devices and systems [122].

### 3.3. PLA Nanocomposites with Natural Nano-Additives

Currently, amongst the key-priorities of the industry and academic sector is the replacement of petroleum-derived and unsafe complexes by the increasing inclusion of natural and green compounds that can be obtained from diverse renewable resources. To achieve promising results, cost-efficient and eco-friendly extraction methods have been designed over the years. Once these green alternatives have been isolated, they are successfully applied to many fields with very assorted aims of utilization such as coagulants, adhesives, dyes, additives, or biomolecules [125]. The main challenge is to develop high-performance polyphenol-reinforced thermoplastic composites, where the use of natural fillers replaces the usual chemical additives with non-toxic ones, not only to improve the final performance but also to increase the desired multifunctionalities (structural, antioxidant, and antibacterial) [126]. Polyphenols comprise an enormous family of secondary metabolites that are stored in vacuoles of vegetal cells such as esters or glycosides. While this family of compounds is vast, they share some mutual properties, such as the formation of coloured complexes with iron salts, oxidation by potassium permanganate in alkaline media, and easy electrophilic aromatic substitution-coupling with diazonium salts and aldehydes [127,128]. Lignin and tannins are considered polyphenols with high molecular weights. Thus, not only do they possess typical features of the polyphenols group, but the presence of a large number of hydroxyls provides them with the ability to create bonds to reach a stable cross-linked association within several molecules, such as carbohydrates or proteins. This unique characteristic differentiates them from the common group of polyphenols.

#### 3.3.1. Lignin

Lignin (Lgn) (Figure 8) is the second most abundant natural polymer after cellulose [129] and is found in every vascular plant on earth [130]. It is a biodegradable, nontoxic, and low-cost macromolecule [131,132] that has generated great interest due to its multi-functionalities, such as UV resistance, bioactive antibacterial, and anti-oxidation activities. Commercially, it is easily obtained as a byproduct in the paper and pulp industry. Its use is also regularly reported in biorefineries and in carbon-fiber manufacturing. Lignin nanoparticles (Lgn-NPs) have been proven to be beneficial reinforcing materials in polymer nanocomposites, as they enhance their thermal and mechanical properties because of the aromatic rings, while [129,131] in the PLA case specifically, lignin nanoparticles’ antibacterial and antioxidant activity as well as their UV protection and reduced water sorption capacity can enhance the final nanocomposites’ performance. Moreover, lignin’s highly branched polyphenolic structure filled with plentiful functional groups allows for its chemical modification and polarity adjustment in such a way that lignin derivatives can be employed in copolymers, composites, and blends for a variety of applications such as innovative phenolic resins, epoxies, adhesives, and in the packaging industry [130]. Presently, the majority of food packaging materials used worldwide are non-biodegradable petrochemical-based plastics, which are responsible for polluting the environment. Approximately 380 million tons of plastic are produced globally every year and around 40% are used in the packaging industry [133]. This statement has forced researchers to widely study the incorporation of natural agents such as Lgn-NPs into biodegradable natural polymers for active packaging. Active food packaging systems involve the direct incorporation of natural bioactive compounds with antioxidant and antimicrobial activity in food packaging materials, which may possibly improve shelf life, ensure the safety of food, and control undesirable quality variations in the food during storage, transportation, and distribution [134]. Bioactive PLA-based composites could potentially replace the currently used plastic films to reduce the menace of environmental pollution and also help in the valorization of wastes from the food industry.

#### 3.3.2. Tannin

The term tannin (TANN) broadly refers to a large complex of biomolecules of a polyphenolic nature (Figure 9). Tannins are found in most species throughout flora, where their functions are to protect the plants or vegetables against predation and might help in regulating plant growth. They are also used for iron gall ink production, adhesive production in wood-based industries, anti-corrosive chemical production, as a uranium-recovering chemical from seawater, and in the removal of mercury and methylmercury from solutions. There are two major groups of tannins, i.e., hydrolyzable and condensed tannins. While hydrolyzable tannins are present in few dicotyledons species, the natural presence of condensed ones is much more abundant; thus, they represent a major and more valuable source of commercial tannins [136].

Generally, tannins are obtained from natural renewable resources, i.e., plants that are the secondary phenolic compounds of plants. More specifically, tannins are either galloyl esters or oligomeric and polymeric pro-anthocyanidins produced by the secondary metabolism of plants, i.e., synthesized by biogenetic pathways [138]. Condensed tannins, composed of flavonoid units, are one of the most abundant and sustainable biopolymers in plants. Their characteristics include antioxidant, antimicrobial, and stabilizing properties and are attractive for use in polymer materials [139]. Studies conducted in the past have shown that tannins exhibit excellent antioxidant and UV-protective properties when incorporated in polyethylene [140], poly(vinyl chloride) (PVC) [141], polypropylene [142], and poly(vinyl alcohol) (PVA) [143]. Until now, the combination of PLA and bio-fillers has been claimed to be an efficient and beneficial method to produce low-cost biocomposites with superior characteristics such as flame retardancy, mechanical performance, thermal stability, and gas-barrier properties. Considering the natural renewable reinforcements based on polyphenolic materials, lignin and tannins have great potential for the preparation of PLA nanocomposites to enhance functionality, especially toward bioactivity, whilst both phenolics act as free radical scavengers and thus as natural antioxidants that are both UV-resistant and bioactive.

In a recent work, Cresnar et al. [144] prepared two series of PLA-KL and PLA-TANN at various contents (0.5%, 1.0%, and 2.5% (*w*/*w*)) by hot melt extrusion to study their antioxidant and antibacterial properties. The results showed the accelerated antioxidant behavior of all PLA-KL and PLA-TANN composites, which increases with the filler content (Figure 10). Furthermore, the KL- and PLA-based TANN showed resistance to *E. coli*, but without a correlation trend between polyphenol filler content and structure. The water contact angle showed that neither KL nor TANN caused a significant change in the wettability, but only a minor improvement in the hydrophilicity of the PLA composites.

Ainali et al. [145] studied the decomposition mechanism and the thermal stability of PLA nanocomposites filled with both biobased kraft-lignin (KL) and TANN in different contents using the melt extrusion method. It was established that the PLA/KL nanocomposites exhibit better thermostability compared to pristine PLA, while the addition tannin had a minor catalytic effect that can decrease the thermal stability of PLA. The calculated Eα value of the PLA-TANN nanocomposite was lower than that of PLA-KL, leading to a considerably higher decomposition rate constant, which accelerated the thermal degradation. Cresnar et al. [146] studied the crystallization and molecular mobility of these composites. It was found that the extremely slow and weak crystallization rate of PLA was improved by the fillers, via the combined effects of providing sites for crystallization and moderately enhancing polymer mobility (chains diffusion). Remarkably, the crystal structure and semicrystalline morphology were found altered in the composites with respect to the neat PLA. Regarding the segmental mobility of PLA, the addition of both tannin and lignin was found to impose moderate changes, explicitly, slight acceleration, narrowing of relaxation times range, and suppression in fragility; however, no systematic changes in the calorimetric or dielectric strengths were detected. Despite the weak effects, it was concluded that TANN is more effective, and this seems to correlate with their smaller size and moderately better distribution as compared to lignin.

Several of these composites have been successfully applied in the field of food packaging, medical devices, textile sectors, and others, owing to the enhanced compatibility between hydrophilic biopolymers and the hydrophobic PLA polymer matrix.

#### 3.3.3. Nanocellulose (NC)

Cellulose is a polysaccharide that exists as a linear chain, consisting of repeating anhydro-D-glucose units covalently linked by β-1,4-glycosidic bonds [147]. Plants create around 75 billion tons of cellulose every year, making this an extremely abundant material. Nanocellulose (NC), prepared by breaking down cellulose fibers, is one such biodegradable, renewable nanofiller (biopolymer) that yields a low carbon footprint and generally refers to nanosized cellulose with a diameter of less than 100 nm and a length of up to several micrometers. It is considered by experts as a highly promising alternative material for petroleum-based products due to its biodegradability, renewability, eco-friendliness, and nontoxicity. It has been extensively used in diverse fields, such as fibers and clothes, the paper industry, optical sensors, the food industry, and pharmaceutics, among many others [148]. The unique characteristics of nanocellulose, such as superior mechanical properties (strength 2–3 GPa), low density (1.6 g cm^−3^), highly specific surface area (200–300 m^2^g^−1^), and low thermal expansion coefficient (1 ppm K^−1^), make it an ideal building block for flexible functional compounds and in outdoor and engineering applications [149]. Except for improvement in electrochemical performance, nanocellulose-based composites with green and abundant raw materials are also beneficial for the reduction of production costs and the construction of environmentally friendly processes.

Lately, NC has been increasingly facilitated in active packaging applications. For cellulose-derived packaging, three types of cellulose are employed, namely, cellulose nanocrystals (CNC), cellulose nanofibrils (CNFBs), and bacterial nanocellulose (BNC). Currently, CNC and CNF are incorporated as reinforcing agents into various biopolymers, such as PLA, for the preparation of green nanocomposites. NC can synergistically work with other materials to improve the barrier and the thermo-mechanical and rheological properties of the nanocomposites. Intramolecular and hydrogen bonding makes cellulose insoluble in almost all solvents as these bonds produce a great strengthening quality [150]. It has been reported that the films produced from NC can be voluntarily reprocessed and recycled into a packaging film without drastically degrading the properties of the same [151]. The most significant reason behind this would be its carbon neutrality, non-toxic nature, recyclability, and sustainability. Attributed to the distinctive properties of NC, tunable surface chemistry, barrier properties, mechanical strength, crystallinity, biodegradability, non-toxicity, and high aspect ratio, it is a growing credible renewable green substrate in food packaging applications [152].

#### 3.3.4. Nano-Biochar (n-BC)

Of the state-of-art biofuels, biochar (BC) has exhibited the potential to complement solid fuel for harvesting bioenergy, while addressing the vital issues associated with the environment. BC belongs to the category of richly carbonaceous items, and it is derived from biomasses such as agricultural and forestry wastes, crops, wood, leaves, municipal sludge, manure, woodchips, and other C-rich materials. Sludge is generated during the wastewater treatment procedure, which produces a solid waste that needs to be treated and disposed. Nevertheless, it is a promising feedstock to produce BC due to its rich carbon content and nutrients such as ammonia [153]. It can be formed through a variety of treatments such as pyrolysis, torrefaction, and gasification at a wide range of temperatures (200–700 °C) and times. Depending on the process conditions, biochar can be produced as either a primary or side product, accompanied by a variable amount of liquid and gaseous streams (such as bio-oil and syngas). Prior to thermal combustion, biomass is dried, the particles are further heated, and volatile substances are released from the solid. The volatile compounds can form permanent gases (such as CO_2_, CO, CH_4_, and H_2_) or condensable organic compounds (e.g., acetic acid and methanol). Subsequent reactions in the gas phase include cracking and polymerization and can therefore alter the entire product spectrum. Three products can be distinguished from the resulting material: permanent gases, one or more liquid phase(s) (water and tar), and a solid residue [154].

The easy availability of feedstock and the inexpensive production of biochar has made it a material of significance for environmental remediation in recent years [155]. BC presents some advantages over other carbon items (single or multi carbon nanotubes, graphene, and activated carbon), such as its highly specific surface area, good stability, abundant functional groups, great carbon stability, and highly porous structure, and thus it has been used in a wide variety of fields, for example, as a pollutant absorbent and for biosensors, fuel cell, and supercapacitors [156]. These properties have expanded its usage in various sectors, ranging from heat and power production, flue gas cleaning, metallurgical applications, use in agriculture and animal husbandry, as a building material, to medical use. Attempting to decrease greenhouse gas emissions, BC has gained increasing popularity lately as a replacement for fossil-carbon carriers in several of these applications [157].

Nano-biochar is naturally formed during the processing of bulk biochar, while its yield is low (e.g., only ~2.0% in peanut shell-derived biochar). A size reduction process is required to increase the content of NPs in biochar. This process can be easily operated via grinding or milling [158]. Figure 11 presents the most common approaches for the preparation of nano-BC from biomasses.

In comparison with bulk BC that is normally measured on a micron or millimetre scale, nano-biochar is defined as biochar with a particle size of <100 nm. However, a wide range of particle sizes has been considered as nano-biochar in the literature, with particle sizes from 0 to 600 nm being considered as nano-biochar. The particle size of nano-BC is highly dependent on the various feedstocks and preparation methods. For the same feedstock, increasing the pyrolysis temperature can lead to a decrease in particle sizes. For instance, increasing the pyrolysis temperature for rice straw from 400 to 700 °C caused a decrease in particle size from 403 to 234 nm using the centrifuging separation method, as reported by Lian et al. [160]. Likewise, when using the same nano-BC preparation route, increasing the pyrolysis temperature from 300 to 600 °C for rice hull feedstock yielded particles with sizes ranging between 190 and 59 nm.

In addition, conventional BC does not bear electrical conductivity. Instead, the biochar with its size reduced on a nano scale with its activation can be a new research trend for formulating easy and stable biosensors and nanocomposites, owing to its greater surface area and its ability to facilitate rapid electron transmission during sensing and biosensing and offer more adsorption sites for target binding [161]. As a carrier, n-BC could facilitate the migration of natural solutes and contaminants, in contrast with the positive effects of bulk BC, such as holding nutrients and immobilizing hazardous chemicals [162].

### 3.4. PLA/Ceramic Nanocomposites

The development of ceramic nanoparticles with improved properties has been studied with much success in several areas such as in synthesis and surface science. Ceramics are defined as solid compounds that are formed by the application of heat and sometimes pressure, comprising at least two elements, provided one of them is a non-metal or a metalloid. The other substance(s) may be a metal or another metalloid.

The properties that these versatile materials exhibit include high mechanical strength and hardness, good thermal and chemical stability, and viable thermal, optical, electrical, and magnetic performance. In general, ceramic components are formed as desired shapes starting from a mixture of powder with or without binders and other additives, using conventional technologies, including injection molding, die pressing, tape casting, gel casting, etc. The sintering of the green parts at elevated temperatures is furthermore essential to reaching densification. Ceramics are a class of biomaterials extensively employed in biomedical devices [163]. Owing to their ability to be fabricated into a variety of shapes, along with their high compressive strength, variable porosity, and bioactive properties in the body, ceramics are widely facilitated as implant materials. The high similarity in the chemical composition of some ceramics such as calcium phosphate with human bone minerals makes them suitable for use as orthopaedic implants (human skeleton, bones, and joints) and dental materials. These materials show excellent bioactivity, high biocompatibility, and excellent osteoconduction characteristics [164].

#### 3.4.1. Bioglass

Of late, bioactive glasses (bioglass, BG) have emerged as potential biomaterials demonstrating interesting applications as bone-cementing materials in prosthetic medical implants and drug delivery systems. Bioglass represents a subgroup of ceramic materials (i.e., based on SiO_2_, Na_2_O, CaO, and P_2_O_5_) that are comparable to native bone mineral components that have osteoconductivity and good mechanical properties. Owing to their lower silica content (<60%), BG materials have interesting bioactivity properties, thereby producing different forms of hydroxyapatite that ultimately stimulate the natural adhesion by means of biological and morphological fixation [165]. BG has been recognized as among the most important bioceramics for bone regeneration, exhibiting high biocompatibility and positive biological effects after implantation. This material can form a carbonated hydroxyapatite surface layer, and is accountable for the strong bonding between BG and human bone.

Nanobioactive glasses (NBGs) are used in combination with biodegradable polymers to enhance their mechanical performance. NBGs exhibit appropriate properties including biocompatibility, controlled biodegradability, ability to release the cells, ability to connect to both hard and soft tissues, and releasing of ions during the degradation process. This group of materials has the potential to be used in tissue repairs and wound healing, and their positive effects on the angiogenesis process have been demonstrated in some reports. The fidelity of NBGs in tissue repair might be attributed to their mineral structure. Accordingly, ions releasing from NBGs can stimulate tissue repair within the human body and therefore accelerate the process of recovery [166]. Furthermore, morphology and particle size influence the bioactivity of the BG. Studies concluded that an increase of the surface area and porosity of BG can dramatically enhance the bioactivity. Nanoscale BG can also increase in vitro bioactivity (HA deposition) in higher quantities than micrometric particles [167].

#### 3.4.2. n-Hydroxyapatite (HAp)

Hydroxyapatite (HAp) (Ca_10_(PO_4_)_6_(OH)_2_) is a bioactive ceramic material, thermodynamically stable in its crystalline state in body fluid and is the principal inorganic constituent of human bone, making it biocompatible with excellent bone healing properties due to its osteoconductive and osteoinductive capacities [168,169]. Natural HAp is typically extracted from biological wastes or feedstocks such as mammalian bone (e.g., bovine, camel, and horse), marine or aquatic sources (e.g., fish bone and fish scales), shell sources (e.g., cockle, clam, eggshell, and seashell), certain plants, and also from mineral sources (e.g., limestone). Synthetic HAp can be fabricated through various methods, including dry methods (solid-state and mechanochemical), wet methods (chemical precipitation, hydrolysis, sol-gel, hydrothermal, emulsion, and sonochemical), and high temperature processes (combustion and pyrolysis) [170]. Nano-hydroxyapatite (nHA) has the small-size effect of a small crystal grain diameter, large interface, high surface free energy and binding energy, and the unique physical and chemical properties of nanomaterials such as a macro quantum tunnelling effect [171].

The brittle nature of HAp makes it unsuitable for most applications when used alone since it does not meet mechanical requirements. The reinforcement of PLA with ceramics such as nanohydroxyapatite (nHAp) is heavily investigated for tissue engineering and bone regeneration applications and has effectively solved many problems such as high brittleness of hydroxyapatite itself and uncontrollable degradation rate. Moreover, HAp in its nanophase has been found to improve osteoblast cell adhesion and long-term functions both in vitro and in vivo [172].

### 3.5. Nanoclays

Among the variety of nanofillers, nanoclays are the oldest and potentially one of the most interesting and versatile ones [173]. Clays are divided into several classes, such as kaolinite, montmorillonite, sepiolite, smectite, chlorite, illite, and halloysite based on their particle morphology as well as chemical and mineralogical composition. Due to their wide availability, relatively low cost, and relatively low environmental impact, nanoclays have been studied and developed for numerous usages. Approximately 30 different types of nanoclays can be found, which depending on their properties are used in different applications [174].

With the rapid growth of nanotechnology, clay minerals are increasingly used as natural nanomaterials. Nanoclays are nanoparticles of layered mineral silicates with layered structural units that can form complex clay crystallites by stacking these layers [175]. An individual layer unit is composed of octahedral and/or tetrahedral sheets. The different structures of nanoclays are basically composed of alternating tetrahedral silica sheets “SiO_2_” and alumina octahedral layers “AlO_6_” in ratios of 1:1 when one octahedral sheet is linked to one tetrahedral sheet as kaolinite or halloysite. In ratios of 2:1, this structure created from two tetrahedral sheets sandwiching an octahedral sheet such as montmorillonite and sepiolite yields the proportion of 2:1:1 (chlorite). It has been reported that the different types of nanoclays affect the properties of PLA/clays nanocomposites [176].

Solvent casting [177], melt blending [178], and in situ polymerization techniques [179] have been extensively used to synthesize nanoclays containing PLA composites. In recent years, nanoclays have been given great consideration due to being able to improve and significantly enhance mechanical and thermal properties, barrier, and flame resistance properties, as well as in their use in the accelerated biodegradation of polymers [173]. However, it has been reported that the different types of nanoclays affect the properties of PLA/clays nanocomposites [176]. Although organic montmorillonite (MMT) [180], bentonite [181], and halloysites nanotubes (HNTs) [182] are the most widely used nanoclays in the synthesis of PLA nanocomposites, the use of other clays with different morphologies, such as sepiolite (a fibrous silicate that has microporous channels running along the length of the fibers) [183] and palygorskite (a fibrous silicate with a needle-like morphology) [184] has also been reported in the literature.

Several studies have shown significant improvement in the properties of PLA because of the addition of nanoclays [185]. At present, nanoclays such as organic montmorillonite (MMT), bentonite, and halloysites nanotubes (HNTs) are being greatly considered as they possess the potential tendency to extensively improve the thermal, mechanical, and functional properties of polymers. However, it has been observed that the properties of synthesized nanocomposites are affected by the amount of nanoclays that is added to the PLA matrix due to the incompatibility between the hydrophobic polymer and the hydrophilic natural nanoclays [186,187]. Specifically, this was observed in the case of MMT nanoclays that present a hydrophilic nature, thus hindering their uniform dispersion in the hydrophobic organic PLA matrix [20]. Studies have reported that the mechanical properties of bio-nanocomposite films improved if a small content of nanoclays was added into the packaging materials. Nevertheless, the mechanical properties of the films declined with a further increase in nanoclay concentration [188,189]. Table 3 overviews the mechanical properties of PLA nanocomposites containing different nanoclays.

## 4. Applications of PLA Nanocomposites

As mentioned above, nanofillers represent an interesting way to extend and improve the properties of PLA in order to prepare high-performance PLA-based nanocomposites. Due to the enhanced properties of PLA nanocomposites, they present the potential as promising applications in the packaging, construction, medical, textile, electronics, solar panel, and agricultural sectors (Figure 12).

### 4.1. Applications of PLA/Metal Oxides

#### 4.1.1. Food Packaging

Metallic-based particles such as MgO, ZnO, SiO_2_ and TiO_2_ have been mainly used in food packaging due to their enhanced antimicrobial properties. In brief, Swaroop et al. [193] produced PLA biofilms reinforced with MgO nanoparticles (up to 4 wt.%) using the solvent casting method on a lab scale for the food packaging sector. Among the prepared biocomposite films, the 2 wt.% reinforced PLA films exhibited the maximum improvement in tensile strength and oxygen barrier properties (up to 29% and 25%, respectively) in comparison to neat PLA films. In general, the studied biofilms were transparent, capable of screening UV radiations and presented superior antibacterial efficacy against the *E. coli* bacterial culture. Taking it a step further, the same group developed for the first time an industrial level melt-processing setup for the preparation of blown PLA/MgO nanocomposite (NC) films for food packaging [194]. In that work, up to 3 wt.% PLA/MgO nanocomposite films were prepared for the investigation of key mechanical, barrier, optical, thermal, and anti-bacterial performance (Figure 13). The tensile strength and plasticity improved by nearly 22% and 146%, respectively, for the 2 wt.% MgO-reinforced films. The oxygen and water vapor barrier properties improved by nearly 65% and 57%, respectively, for the 1% material. For the 1 wt.% NC films, around 44% of the *E. coli* bacteria were killed after a 24 h treatment, indicating that these nanocomposites could be used as a sustainable alternative for petroleum-based packaging film materials.

Chong et al. [195] conducted a literature report regarding the fabrication of PLA-ZnO nanocomposites with enhanced antibacterial properties. Although ZnO’s antibacterial mechanisms are still under thorough investigation, ZnO NPs have been found to be active against biofilm formation due to the generation of hydroxyl radicals, thus increasing the antibacterial activity of commonly used antibiotics. They highlighted that their multi-functionality and versatility enable their use in antibacterial, UV-absorption, and photocatalytic applications such as food packaging products.

Zhang et al. [50] developed PLA/ZnO NPs electrospun membranes with enhanced antimicrobial activity and UV blocking for food packaging applications. The optimal performance in terms of antimicrobial activity was achieved with 0.5 wt.% ZnO NPs. Specifically, the inhibition zones of *Escherichia coli* and *Staphylococcus aureus* strains were 15.02 ± 0.18 mm and 14.74 ± 0.06 mm, respectively, concluding that this composite is a suitable alternative as a food packaging material that provides both an antibacterial effect and serves as a UV light barrier.

TiO_2_ is a material with bactericide activity that is improved in the presence of light [196]. Baek et al. [197] modified the surface of TiO_2_ with oleic acid (OA) to make it nonpolar and thus improve its compatibility with PLA and PLA films that were prepared using solvent casting. The surface modification with oleic acid improved the dispersion of the nanoparticles in PLA matrices more so than unmodified TiO_2_, as verified by SEM micrographs. Oxygen permeability and water vapor permeability of 1% OT-PLA were reduced by 29% and 26%, respectively, when compared to pure PLA. OT-PLA had higher transparency than T-PLA but provided better light blocking in UVA and UVB regions than PLA. Feng et al. [34] investigated the antibacterial activity of PLA/TiO_2_ nanocomposite films and nanofibers against *E. coli* and *S. aureus* using two different lighting conditions: a UV-A (360 nm) irradiation and a fluorescent lamp source. The pristine PLA presented almost no antibacterial properties, while with increasing amounts of TiO_2_ NPs, and the antibacterial strength of the nanocomposite films rose evenly. The maximum inhibition ratio was observed at a TiO_2_ content of 0.75 wt.% for both NPs and films. Segura- González et al. [93] also examined the antibacterial activity of PLA/TiO_2_ nanocomposites against *E. coli* (DH5α). The basic conclusion was that the presence of TiO_2_ NPs led to a decline in the biofilm development and in the bacterial growth of *E. coli*. These results may be attributed to an immediate impedance on bacterial metabolisms. Furthermore, bacteria size on the nanocomposites incorporating nanoparticles with a diameter of 21 nm was even smaller than one grown on the nanocomposites incorporating the <100 nm nanoparticles. This observation may relate to the surface-to-volume ratio of the NPs, which might alter the oxidative catalytic performance of TiO_2_ in the direction of the organic material related to both the EPS (extracellular polymeric substance) and the bacteria themselves.

An alternative material in the field of nano-food packaging was also proposed by Batool et al [198]. In their work, ZnO NPs hexagonal in shape and 36.5 nm in size at concentrations of 0.4% and 4% (*w*/*w*) were incorporated into a PLA matrix using the solution casting technique. ZnO-NPs were prepared by *aloe barbadensis’* leaves extract and their addition in PLA enhanced stability, elongation, and film thickness. The antimicrobial assay of fruit and biofilm against *E. coli* and *S. aureus* bacterial strain results confirmed that nano-ZnO-based packaging film improved the shelf life and food quality of Vitis vinifera fruit up to two weeks at 40 °C.

#### 4.1.2. Medical Applications

Except for food packaging, PLA/Metal oxide nanocomposites can be also used in medical applications such as in tissue engineering, wound healing, and drug delivery systems [199,200,201].

In a very recent study, Grande-Tovar et al. [199], prepared membranes based on polycaprolactone (PCL) and PLA incorporated with ZnO-NPs, glycerol (GLY), and tea tree essential oil (TTEO) for tissue engineering applications. It was found that the incorporation of the ZnO-NPs and TTEO accelerated the degradation process of the PLA/PCL matrix. All membranes exhibited biodegradability and biocompatibility after 60 days of study, allowing a healing procedure to occur with the recovery of tissue architecture and hair without the occurrence of an aggressive immune response. The membranes are fragmented and reabsorbed by inflammatory cells during the resolution process (Figure 14). Simultaneously, the implantation zone was replaced by connective tissue with type III collagen fibers, blood vessels, and some inflammatory cells that continue the reabsorption process.

In an attempt to prepare alternative materials for bone implant applications, Nonato et al. [202] molded PLA/ZnO nanofibers (1 wt.%) using the solvent-cast three-dimensional (3D) technique. Conditions were adapted to imitate a mechanical fatigue test at human body temperature–cyclic stress in an isotherm at 36.5 °C. The obtained results indicated that for temperatures above 30 °C, the storage module of PLA/ZnO nanocomposites was higher when compared to pure PLA, and in the fatigue test, PLA/ZnO withstood more than 3600 cycles, while pure PLA failed after an average of 1768 cycles. Once again, the excellent antimicrobial activity of ZnO nanocomposites was verified against numerous bacterial strains such as *Staphylococcus aureus*, *Salmonella*, *E. coli*, as well as *Candida albicans* yeast.

In another work, Rashedi et al. [203] prepared PLA/ZnO nanofibrous nanocomposites loaded with tranexamic acid (TXA), which was prepared using the electrospinning process for wound healing patches. The obtained nanofibers exhibited a small pore size that enables the appropriate permeation of atmospheric oxygen to the wound. An in vitro cytotoxicity assay of the nanofibrous mats showed the wound dressing material did not cause any harmful effect upon the human dermal fibroblast cells. Antibacterial studies against Gram-negative *Escherichia coli* and Gram-positive *Staphylococcus aureus* revealed a 75% and 98% reduction in colonies of the bacterial strains, respectively. In vivo tests on mice models evidently demonstrated that the PLA/ZnO/TXA nanofibrous nanocomposites dressing enhanced the wound healing process (Figure 15). These results encourage the use of the suggested nanocomposite as a helpful wound dressing where rapid wound healing and proliferation of skin cells are mandatory.

Monadi et al. [204] incorporated SiO_2_ in the PLA matrix at different concentrations (1, 3, and 5 wt.%) to produce a PLA nanocomposite film with enhanced antimicrobial properties for food packaging applications. They found high antimicrobial activity against Escherichia coli and Staphylococcus aureus, whereas the water vapor permeability and oxygen transfer rate presented the lowest values, decreasing by 56.9% and 55.2%, respectively, compared to neat PLA.

Moreover, Wang et al. [205] grafted PLA chains with a small amount of functionalized SiO_2_ (f-SiO_2_). Because of the improvement in dispersion and interfacial interaction in the PLA matrix, the f-SiO_2_ illustrated an effective reinforcing and toughening effect for PLA, where the elongation at break, tensile strength, and impact toughness of PLA nanocomposite improved by 47.8, 14.9, and 30.3%, respectively, compared to PLA neat. Additionally, the degree of PLA crystallinity was significantly enhanced by the added f-SiO_2_.

The objective of a recent study [206] was to prepare a series of poly(DL-lactic acid) (PDLLA) nanocomposites with four different amounts of silica (SiO_2_) nanoparticles (2.5, 5, 10 and 20 wt.%), following a new two-step synthesis route: ring opening polymerization of DL-lactide and polycondensation. According to intrinsic viscosity measurements, the average molecular weight of PDLLA (Mn ≈ 38,097 g mol^−1^) decreased with increasing SiO_2_ content. SEM images confirmed the fine dispersion of SiO_2_ nanoparticles inside the polymer matrix due to the interactions between ester groups of PDLLA and surface silanol groups of SiO_2_. In addition, they confirmed that thermal stability was increased by the addition of SiO_2_ in the PLA matrix.

#### 4.1.3. Environmental Applications

In the last 30 years, the photocatalytic efficiency of nanosized metal oxides at degrading organic contaminants in the air and water has resulted in their utilization in several environmental applications [207]. These include photocatalytic hydrogen production via water splitting, degradation of organic pollutants in wastewater photocatalytic self-cleaning, bacterial disinfection photo-induced super hydrophilicity, as well as in photovoltaics and photosynthesis [99].

Bobirică et al. [208] prepared novel photocatalytic membranes based on PLA/TiO_2_ hybrid nanofibers deposited on fiberglass supports for the removal of ampicillin from aqueous solutions via the electrospinning technique. The fiberglass fabric plain woven-type membrane showed the highest efficiency of ampicillin removal from the aquatic solutions. However, the degree of mineralization of the aqueous solution is kept low even after two hours of photocatalysis, owing to the degradation of PLA from the photocatalytic membrane. Li et al. [91] synthesized via the sol-gel method TiO_2_ nanoparticles (6.3–11.1 nm) using titanium tera-isopropoxide (TTIP) as the hydrolysis material for the reinforcement of PLA films by casting. The photocatalytic activity of films was verified by methyl orange (MO) solution degradation under UV irradiation. Photodegradation of PLA due to TiO_2_ NPs was associated with the development of carbon-centered radicals and acceleration of interfacial polymer chains gap by active oxygen species. For neat PLA, after 12 h of UV radiation, the degradation of MO observed was 55%, while the film containing 0.6 wt.% TiO_2_ reached a photocatalytic degradation efficiency of 99% after 12 h of UV irradiation.

By using mild and environmentally friendly synthetic conditions, Gupta et al. [209] developed photocatalytic nanocomposites composed of porous PLA microparticles via oil/water emulsion with incorporated anatase TiO_2_ NPs for the sorption and UV-triggered degradation of organic compounds. The authors demonstrated that the sorption capacity, dye degradability, and composite disintegration can be controlled by altering the PLA microparticles’ porosity and the distribution of incorporated titania NPs. Both types of PLA/TiO_2_ composites removed rhodamine 6G from water (up to 60% of the initial amount in six hours of UV exposure time) unlike negligible dye removal observed for TiO_2_-absent PLA particles. The authors suggested that these composite microsponges may be facilitated as nontoxic photocatalytic materials for the efficient environmental clean-up of contaminated water.

The photocatalytic degradation of a mixture of cytostatic drugs has been studied using immobilized TiO_2_ on polymer films (PET and PLA) [207]. Both studied nanocomposite materials have been proven to be effective. However, higher degradation rates were achieved in the presence of PET-TiO_2_.

#### 4.1.4. Other Applications

It has also been reported that PLA/metal oxide nanocomposites are alternative candidates for air filters applications as well as in electronic and automotive applications (Table 4).

Petousis et al. [210] reported the effect of Al_2_O_3_ NPs at four different filler loadings, as a reinforcing agent of PLA using FFF for a series of potential applications. A positive reinforcement mechanism was observed at all filler loadings, while the mechanical percolation threshold with the maximum increase of performance was found between 1.0–2.0 wt.% filler loading (2.0 wt.% in PLA, and a 40.2% and 27.1% increase in strength and modulus, respectively). Sukhanova et al. [211] investigated the incorporation of epoxidized aluminum oxide nanofibers (AONF) nanofiller in improving the characteristics of composite polymer films based on biodegradable PLA by solution casting. It was shown that the incorporation of AONF results in an enhancement in the mechanical properties of neat PLA. Kangali et al. [212] prepared PLA/boron oxide nanocomposites by the solution casting method as alternative candidates for electronic, packaging, and automotives industries. The particle size of boron oxide was reduced to a nanosize using SPEX ball mill and boron oxide nanoparticles (BONPs) that were functionalized with oleic acid, [3-(trimethoxylsilyl) propylmethacrylate] (MPTMS) and triethoxyvinylsilane (VS). Mechanical and flammability tests showed that the nanosized and surface functionalized BONPs enhanced the crystallinity and mechanical performance of PLA/BONPs. The crystallinity value increased from 13.68% to 32.55% with the increasing functionalized BONPs in the PLA matrix. Likewise, the tensile strength of the nanocomposites increased from 41.25 ± 0.80 MPa to 51.12 ± 2.54 MPa with the increasing functionalized BONPs.

Wang et al. [213] prepared hybrid PLA/TiO_2_ NPs fibrous membranes presenting good antibacterial activity via the one-step electrospinning technique for air filter applications. Filtration performance tests (Figure 16) indicated that fibers with a high surface roughness, large specific surface area, and large nanopore volume greatly improved the particle capture efficiency and facilitated the penetration of airflow. From the overall tests conducted, it was concluded by the authors that the PLA/TiO_2_ fibrous membrane loaded with 1.75 wt.% TiO_2_ NPs prepared at a relative humidity of 45% was the optimal material, since it showed excellent filtration efficiency (99.996%) and a relatively low pressure drop (128.7 Pa), as well as a high antibacterial activity of 99.5% against *Staphylococcus aureus*.

**Table 4 polymers-15-01196-t004:** Applications of PLA/metal-oxide nanocomposites in various sectors.

Material	Application	Properties	References
PLA-ZnO	Packaging, tissue engineering, wound healing, drug delivery, disposable electronics	Dielectric properties, nti-inflammatory and antibacterial activity, biocompatibility	[195,199,203]
PLA-TiO_2_	Packaging, air filters, tissue engineering, wound healing, electronics	Total anti-UV protection, optical and antibacterial properties, nanocomposites with higher kinetics of crystallization, photodegradability, etc.	[94,214,215]
PLA/Al_2_O_3_ and PLA/BO	Electronics, packaging and automotives industries	Anti-inflammatory properties and non-toxicity against fibroblast L929 cells	[210,211]

### 4.2. PLA/Carbon-Based Nanocomposites

#### 4.2.1. Thermal and Electrical Applications

It has been demonstrated that carbon-based materials including carbon black, activated carbon, graphite, CNFs, and CNTs are very effective fillers in improving the thermal insulation performance of polymer foams by reducing the radiative heat transfer [132]. Wang et al. [216] reported an eco-friendly and versatile process to fabricate ultra-low-threshold and lightweight biodegradable PLA/MWCNTs foams with segregated conductive networks for high-performance thermal insulation and electromagnetic interference shielding to meet the rising request for high-performance multifunctional materials in sustainable development. Owing to the unique structure of the microporous PLA matrix embedded by conductive 3D MWCNTs networks, the lightweight porous PLA/MWCNTs with a density of 0.045 g/cm^3^ possess a percolation threshold of 0.00094 vol%, which was the minimum value reported at the time. Moreover, the developed material showed excellent thermal insulation performance with a thermal conductivity of 27.5 mW·m^−1^·K^−1^, significantly lower than the best value of common thermal insulation materials.

A simple, CO_2_-based and eco-friendly yet effective foaming methodology for fabricating ultra-low-density PLA/CNTs nanocomposite foam for thermal insulation application was reported by Li et al. [217]. With the gradual incorporation of CNTs, three kinds of networks were generated in PLA/CNTs nanocomposites and had a distinct reinforcement influence on their melt viscoelasticity. The storage modulus of PLA/CNTs nanocomposites were three orders of magnitude higher in contrast to neat PLA. Interestingly, relative to abnormal DSC, a double melting peak phenomenon appeared in the high-pressure DSC curves of various PLA specimens. Biodegradable PLA/CNTs nanocomposite foam was successfully fabricated at a foaming temperature (T_f_) of 121 °C with a super-high volume expansion ratio (VER) of 49.6 times. The pore size, pore density, and VER of diverse PLA specimens were tuned and regulated efficiently by the content of CNTs and foaming temperature. Finally, the authors underlined that these results offer a promising strategy for developing other thermoplastic polyester foams with ultra-high VER to obtain some unique functional attributes.

Sanusi et al. [218,219,220] incorporated hybrid nanofillers containing MWCNTs and montmorillonite (Mt) at concentrations of 0.5, 1 and 2 wt.% through a two-step procedure of solution and melt mixing, to create reinforcing fillers in the PLA matrix. The physicochemical studies indicated a strong hybrid–polymer interaction. The structural analyses confirmed the synergy between the nanoclay and the carbon nanotubes in reaching a homogeneous dispersion of Mt/MWCNT NPs throughout the PLA. Furthermore, the addition of a low mass fraction of the studied hybrid nanofillers led to a more than 24% and 45% enhancement in the tensile strength and elastic modulus of PLA, respectively. Finally, at low loading (0.5 wt.%) of the Mt/MWCNTs nanofiller, the performance of the thermal degradation was improved, while an increased amount of added nanohybrid implies a well-established percolation network in the nanocomposite, which might serve as a good thermal conductive material.

Graphene and its different types also have excellent electrical and thermal properties. Kalinke et al. [221] reported the comparison of the electrochemical properties of 3D-printed PLA-graphene electrodes (PLA-G) under different activation conditions and through different processes. The sensor showed good repeatability and reproducibility and the electrodes were successfully applied to DA determination in synthetic urine and human serum, showing good recovery, from 88.8 to 98.4%. Thus, the activation methods were vital for the improvement in the 3D PLA-G electrode properties, allowing graphene surface alteration and electrochemical enhancement in the sensing of molecular targets.

In another work, the introduction of exfoliated graphene into the PLA matrix produced conductive PLA/graphene nanocomposites via the solution casting method [222]. Among the concentrations of the filler (0.5, 1.5, 1.7, 2, 2.5, and 3 wt.%) added to PLA, the content of 1.7 wt.% graphene was the optimal choice since a significant drop in impedance (10^5^ Ω) compared to neat PLA (10^11^ Ω) was noticed. As the graphene loading was increased, the impedance gradually reached the reduced value of 10^4^ Ω. In addition, the specific nanocomposites suggest the reusability of the films for at least five cycles and the capability to reproduce these results in nature.

Finally, Spinelli et al. [223] evaluated the temperature effect on the thermophysical properties of PLA nanocomposites reinforced with two different weight percentages (3 and 6 wt.%) of GNPs. At the lowest temperature (298.15 K) measured, an enhancement of 171% was observed for the thermal conductivity compared to the unreinforced matrix due to the addition of 6 wt.% GNPs, whereas at the highest temperature (372.15 K) such an improvement was about 155%. Moreover, similar to the glass transition (expected at ~333 K), the thermal diffusivity decreased with increasing temperatures.

Masarra et al. [224] printed electrically conductive PLA/PCL composites using the FFF that were mixed with different contents of GNPs. The electrical resistivity results using the four-probes method revealed that the 3D-printed samples featuring the same graphene content are semiconductors. Varying the printing raster angles also exerted an influence on the electrical conductivity results. The electrical percolation threshold was found to be lower than 15 wt.%, whereas the rheological percolation threshold was found to be lower than 10 wt.%. Furthermore, the 20 wt.% and 25 wt.% GNP composites were able to connect to an electrical circuit. Lastly, an increase in the Young’s modulus was shown with the percentage of graphene. In a PLA/poly(methyl methacrylate) (PMMA) blend (ratio of 40:60 *w*/*w*), two types of graphene were added to prepare nanocomposites via solution-mixing blending, graphene nanoplatelets (GNPs), and acid-modified graphene (FG) [225]. It was observed that the addition of FG to the PLA/PMMA blend indicates a lower percolation threshold and higher electrical conductivity compared with the blend filled with GNPs due to FG’s better homogenous dispersion and uniform distribution in the polymer phases.

Silva et al. [226] synthesized biodegradable blends of PLA with poly(3-hydroxybutyrate*-co-*3-hydroxyvalerate) (PHBV), enriched with CNTs for electrical, electromagnetic, and military applications. More specifically, the PLA/PHBV blend (80/20) and PLA/PHBV blend-based nanocomposites with 0.5 and 1.0 wt.% CNT were produced. SEM analysis showed that PHBV was homogeneously dispersed in the PLA matrix and that CNTs are preferably dispersed in the PHBV phase. The CNT acted as a nucleating agent for the crystallization of PHBV and had no effect on the thermal stability of the nanocomposites. The addition of 1.0 wt.% CNT resulted in better electrical properties (2.79 × 10^−2^ S/m) and an excellent result as an electromagnetic interference shielding material (attenuation of approximately 96.9% of the radiation in the X-band).

The hydrolytic degradation of PLA/poly(ethylene oxide) (PEO) blends and their CNTs nanocomposites were also investigated by Zare et al. [227]. The fine dispersion of CNTs in the nanocomposites also demonstrated the strong interfacial interactions between the polymer and CNTs. The samples showed different melting peaks, suggesting that PLA and PEO are immiscible. The thermal decomposition of PEO was accelerated by the addition of CNTs in all samples because of the better heat transfer to the dispersed phase in the nanocomposites.

The capability of fullerenes to form supramolecular complexes with different types of molecules has been featured in applications mainly in the development of organic photovoltaic cells. The topic of supramolecular complexes of fullerenes with macromolecules such as PLA was explored by Cataldo et al. [228]. Spectrophotometric tests revealed that a new band in the UV, i.e., at 230 nm, was assigned to the charge-transfer interaction between PLLA and C60, whereas the former acts as an electron donor through the ketones of the ester group and the latter as an electron acceptor being an electron-deficient olefin. From this study, it was confirmed that PLLA and C_60_ are in close contact with each other, suggesting a kind of host–guest interaction where the helical conformation of PLLA is able to induce the optical activity to the achiral C_60_ electronic transitions.

#### 4.2.2. Biomedical Applications

CNTs have been also viewed as an interesting additive to improve the performance of 3D-printed PLA parts for biomedical applications, owing to the biocompatibility and biodegradability of PLA [229]. Magiera et al. [230] compared the degradation behavior of electrospun-nanofibers destined for biomedical applications, obtained from neat PLA and PLA/CNTs composites in aquatic environments. PLA and PLA/CNT composite nanofibers underwent swelling and partial degradation during incubation due to the penetration of water into the polymer matrix. An increase in the fibers’ diameters occurred, although the overall porosity of the composites remained unaltered. Changes in the mechanical properties of the composite mats were higher than those observed for pure PLA mats. After 14 days of incubation, the samples that retained 47 to 78% of their initial tensile strength were higher than pure PLA samples. Morphological changes in pure PLA nanofibers were more dynamic than in composite nanofibers, which indicates their higher stability under the experiment conditions, while no significant changes in the crystallinity, wettability, and porosity of the samples were observed.

Also employing the electrospinning method, Zhang et al. [231] incorporated MWCNTs and doxorubicin (DOX) into the PLLA nanofibers for localized cancer treatment by the combination of both chemo- and thermotherapy. The multifunctional fibers presented increased cytotoxicity both in vitro and in vivo by the combination of photothermal-induced hyperthermia and chemotherapy with DOX. In total, photothermal treatment and chemotherapy were successfully integrated into one single system using the electrospun DOX/MWCNTs-loaded PLLA nanofiber mats. The fiber mats realized local drug delivery and hyperthermia and at the same time and showed significantly enhanced anti-tumor efficacy for this combination therapy strategy. The low energy of irradiation did not only induce cancer cell death via the hyperthermia effect, initiating the burst release of DOX from the fibers due to the relatively low Tg of PLLA, but also significantly increased the temperature of the fiber-covered tumor site. The latter resulted in an enhanced restraining effect on tumor growth and minor side effects on other normal organs.

Lee et al. [232] prepared micro needle patterns (Figure 17) of PLA/MWCNTs using the injection molding method and studied the effects of MWCNTs on crystallization, thermal behavior, and the replication and surface properties. It was concluded by the authors that the processing parameters greatly affect the replication and surface properties of the micro injection-molded PLA/CNT nanocomposites. Specifically, an analysis of the thermal behavior and crystallinity indicated that the MWCNTs promoted the unique α’ to α crystal transition of PLA, resulting in an enhancement of surface modulus and hardness. In addition, the MWCNTs increased the activation energy for thermal degradation of PLA due to the physical barrier effect. The replication quality (ratio higher than 96%) of the micro-features in the PLA/MWCNTs nanocomposites was accomplished by elevating the injection speed (120 mm/s) and holding pressure (100 MPa), which enhances the polymer filling ability within the micro cavity.

In the work conducted by Vidakis et al. [233], an industrially scalable method was developed for the preparation of multifunctional nanocomposite filaments suitable for multiple industrial applications, such as sensors fabrication, health monitoring devices, medicine etc. Briefly, the PLA polymeric matrix was enriched with MWCNTs at 0.5, 1.0, 2.5, and 5.0 wt.% filler loadings to fabricate novel materials using 3D-printing technology. The performed tests showed that the addition of MWCNTs at loadings higher than 1 wt.% significantly improved the mechanical properties and rendered the nanocomposites electrically conductive. Furthermore, the 5 wt.% loading showed mild antibacterial activity against *E. coli* and *S. aureus* colonies.

The combination of C60 and PLA could have possible medical application in the preparation of a new type of suture or stent with various medical functions. Keeping that in mind, Thummarungsan et al. [234] reported the preparation of electroactive samples based on the dibutyl phthalate (DBP)-plasticized PLA and fullerene (C60) produced via the solution casting method for biomedical applications. All PLA composites presented fast and reversible responses when subjected to electrical stimulus. The C60/PLA/DBP composite at a concentration of 1.0% *v*/*v* showed the highest storage modulus response up to 23.51 × 10^5^ Pa under the 1.5 kV mm^−1^ electric field. Thus, the authors claimed that the electrically responsive PLA composites prepared in their work that have a short response time and a high bending deformation were demonstrated to be promising biobased materials for actuator applications.

Chen et al. [235] synthesized for the first time PLLA composites with a C60 tetragonal single crystal (C60TSC) through facile evaporation of the CHCl_3_ solution containing PLLA and C60. TGA tests showed that in the second step of mass loss, the thermal decomposition peak temperature of PLLA in the PLLA/C60TSC composite rises to 352.6 °C, 29.7 °C higher than that of pure PLLA, indicating that the resultant C60TSC efficiently enhanced the thermal stability of PLLA. Overall, the incorporation of C60TSC also led to the formation of PLLA composites that have a higher melting temperature and higher cold crystallization temperature, as well as a higher glass transition temperature and lower relative crystallinity (Xc) than pure PLLA.

In another study, inorganic fullerene (IF)-like tungsten disulphide (WS_2_) nanoparticles, ranging between 0.1 and 1 wt.%, from layered transition metal dichalcogenides (TMDCs), were successfully introduced into a PLLA polymer matrix by Naffakh et al. [236]. The main objective of this research was to generate novel bio-nanocomposite materials through an advantageous melt-processing route. It was discovered that the incorporation of increasing IF-WS_2_ contents led to a progressive acceleration of the crystallization rate of PLLA. The morphology and kinetic data demonstrated the high performance of these novel nanocomposites for industrial applications.

Li et al. [237] developed a C60 L-phenylalanine (phe) derivative attached with PLA (C60-phe-PLA) for the preparation of injectable Mitoxantrone (MTX) antitumor drug multifunctional implants. C60-phe-PLA was self-assembled to form microspheres (Figure 18a) consisting of a hydrophilic antitumor drug (MTX) and a hydrophobic block (C60) using the dispersion–solvent diffusion method. The results obtained from the performed investigation were very promising for the future treatment of solid cancer tumors. Specifically, the self-assembled microspheres showed a sustained release pattern within 15 days of in vitro release studies, as presented in Figure 18b. According to the tissue distribution of C57BL mice after intratumoral administration of the microspheres, the MTX was mostly distributed amongst the tumors, and was hardly detected in the heart, liver, spleen, lungs, and kidneys. Microspheres provided high antitumor efficacy with negligible toxic effects on normal organs, owing to their significantly increased MTX tumor retention time, low MTX levels in normal organs, and strong photodynamic activity of PLA-phe-C60. In summary, the authors stated that the prepared C60-phe-PLA microsphere could serve not only as a powerful photo dynamic therapy but also as a sustained-release drug delivery vehicle, suggesting that there is great potential for this formulation for local cancer treatment.

#### 4.2.3. Structural Applications (Mechanical-Thermal Properties Enhancement)

CNTs can enhance the crystallization rate and mechanical properties of PLA. With that knowledge, Bortoli et al. [238] functionalized CNTs with an HNO_3_ solution to create defects and incorporate oxygen functional groups on the CNTs surface for the assessment of the thermal and mechanical properties of 3D-printed PLA/CNT nanocomposites. The results suggested that the functionalized CNTs (f-CNTs) displayed an improved dispersion in the matrix and acted as effective nucleating agents for PLA crystallization, when compared to the use of commercial CNTs (c-CNTs). The addition of just 0.5 wt.% f-CNT was adequate to yield a significant increase in the mechanical strength of the 3D- printed parts (from 29.4 ± 0.7 MPa for PLA/c-CNT to 41.6 ± 1.4 MPa for PLA/f-CNT) and provided better interfacial adhesion between 3D-printed layers, maintaining the thermal stability of the nanocomposites (Figure 19). DMA analysis indicated a significant increase in the storage modulus (~43% at 37 °C) when f-CNTs were used as a reinforcement (Figure 20). Consequently, both mechanical and thermal properties of 3D-printed PLA/CNT nanocomposites were significantly enhanced when f-CNTs were employed, also indicating that percentages of less than 1 wt.% of this additive are required.

In a recent study by Yang et al. [56] a filament based on PLA/CNTs composites was prepared for the fused deposition modeling (FDM) process. The effects of the CNT content on the crystallization-melting behavior and melt flow rate were tested to investigate the printability of the PLA/CNT. The results demonstrate that the CNT content has a significant influence on the mechanical properties and conductivity properties. The addition of 6 wt.% CNT resulted in a 64.12% increase in tensile strength and a 29.29% increase in flexural strength. The electrical resistivity varied from approximately 1 × 10^12^ Ω/sq to 1 × 10^2^ Ω/sq for CNT contents ranging from 0 wt.% to 8 wt.%.

Lately, PLA-graphene nanocomposites have been extensively fabricated using additive manufacturing such as 3D printing due to the attractive properties offered by both materials. However, in composites, the main challenge is to understand the graphene properties’ transfer from the nanoscale to the macroscale. Camargo et al. [239] investigated the effect of the variation of the infill and layer thickness parameters on the mechanical behavior of 3D-printed materials. Due to the layered production process, 3D-printed parts exhibit anisotropic behavior. The data obtained showed that the mechanical performance improved with the enhancement of the layer thickness and infill density parameters, while impact energy decreased as the infill increased. Caminero et al. [240] reported that the printed PLA-graphene composite samples showed the best performance in terms of surface texture, tensile strength, and flexural stress in comparison to pristine PLA. Nevertheless, the impact strength of the PLA-graphene composites was reduced by 1.2–1.3 times compared to that of the un-reinforced PLA matrix. Furthermore, the Plaza group examined the coupled effect of the cyclic loading amplitude and frequency on the fatigue behavior of materials (especially composites) and the induced self-heating phenomenon [241]. It was shown that at high values of frequency and applied stress during the first stage of low cycles fatigue, composites exhibit an overall fatigue response mainly governed by induced thermal fatigue (ITF). Accordingly, the mechanical fatigue (MF) nature becomes predominant during the second stage before the failure. For low frequency and applied amplitude, no significant self-heating phenomenon has been observed.

In the study conducted by El Magri et al. [242], the combined effect of process parameters, loading amplitude, and frequency on the fatigue behavior of the 3D-printed PLA-graphene specimens were analyzed. The obtained experimental results highlighted that fatigue lifetime clearly depends on the process parameters as well as the loading amplitude and frequency. In addition, when the frequency was 80 Hz, the coupling effect of thermal and mechanical fatigue caused self-heating, which decreases the fatigue lifetime.

Garcia et al. [243] compared geometric properties such as the dimensional accuracy, flatness error, surface texture, and surface roughness of a 3D-FFF-printed PLA-GNP reinforced matrix. The results showed that the dimensional accuracy was mostly affected by the build orientation, which showed an increase in the layer area on the X–Y plane and the highest dimensional deviation owing to the longer displacements of the extruder accumulating positioning errors. Therefore, it is clear that PLA-graphene filaments improved the mechanical, electrical, and thermal properties without losing their geometric quality.

Ghani et al. [244] evaluated the mechanical properties of PLA reinforced with GNPs at a concentration range of 0.1 wt.% to 1 wt.% via the meld-blending technique and then injection molding. At 0.3 wt.% filler, the tensile strength increased, reaching the maximum value of 50.3 MPa, owing to improved dispersion inside the PLA matrix. The elongation at break is enhanced with the further addition of the filler, resulting in the highest value of 2.32% with 0.5 wt.%, although with 1 wt.% graphene, this property decreased because the excess of the filler limits the motion of the polymer chains and makes the nanocomposite more brittle.

Nanocomposites of PLA reinforced with graphene oxide nanosheets using the electrospinning method also enhanced the mechanical properties [245]. The addition of GO nanosheets strengthened the PLA fiber mat matrix with the ultimate tensile strength, reaching the value of 4.4 ± 0.46 MPa from 1.7 ± 0.43 MPa of neat PLA fiber mats. The modulus was also improved from 0.53 ± 0.07 MPa of neat PLA to 0.95 ± 0.24 MPa due to the GO nanosheets, which demonstrates a strong interfacial interaction with the matrix. Nevertheless, elongation at break was decreased from 31% of neat PLA to 17.7% of PLA/GO fiber mats.

The majority of the current research on additive manufacturing has been concentrated on improving the mechanical properties, such as strength and stiffness, of polymer composites. Papon et al. [246] reinforced PLA with CNF and investigated the effect of filler concentrations (0.5 wt.% and 1 wt.%) on the composites’ properties. Mechanical studies indicated an increase of 12% in the Young’s modulus at the concentration of 1 wt.% of the filler. Although the failure strength demonstrated an increase at 0.5 wt.% CNF, by increasing the content up to 1 wt.%, their value decreased due to agglomeration of the filler, low CNF aspect ratio, and poor interfacial bonding between the polymer matrix and the filler.

Malafeev et al. [247] investigated the mechanical properties of PLA using vapor grown carbon nanofibers (VGCF) as a filler via a twin-screw micro-extruder. An improvement of 20% in strength was observed at high-temperature orientational stretching of the nanocomposite by four times, regardless of the concentration of the filler, while the elastic modulus and deformation at break were at the initial level of the pristine fibers. In contrast, stretching by six times caused a decrease of 15% in strength and elastic properties and a possibility of existing microcracks in polymer fibrils was suggested. Dave et al. [248] reinforced PLA using a carbon nanofiller (0.1 wt.%) in the form of carbon quantum dots to create composite scaffolds through melt blending and underwent a 7 min plasma treatment for the improvement of cell adhesion and growth, obtaining superhydrophilicity and cell proliferation, which had increased by 70% compared to untreated ones. An increase of 24.1% in tensile strength was noticed compared with pure PLA due to the enhanced surface area of intercalated nanoparticles and effective load transfer and dissipation at the interface of carbon dots with PLA.

Nanocomposites of PLA reinforced with carbon NPs via compression molding technique exhibited an improved tensile strength due to good dispersion of NPs in the PLA matrix, the formation of hydrogen bonds and the increasing composition of the filler, ranging from 2 wt.% to 8 wt.% [249]. Regarding the Young’s modulus, even though there was a 25% increase, only with the addition of 8 wt.% of the filler, a decrease was observed due to additional quantity of nanocarbon that hinders the bonding of some particles with the PLA matrix.

Improved electrical resistivity, compared to a solid PLA filament, in the range of 64 ± 25 to 1.4 ± 0.48 Ω m for 6–15 wt.% PLA/carbon NPs filaments, was reported by Potnuru et al. [250]. In a further step, 3D samples using the abovementioned filaments with 25 MPa mechanical strength and 20 ± 10 Ω m electrical resistivity were produced. The authors highlighted that this method can be facilitated to build humanoid robot structures with electrical circuitry. Enhanced electrical conductivity (3.76 S/m) was also claimed by Jain et al. [251], owing to chemical bonding between PLA and the nanofiller, during the fabrication of PLA/carbon nanopowder filaments for 3D printing. Solution blending was used for PLA-NC nanocomposite fabrication and melt extrusion was employed to make cylindrical filaments.

Carbon-based nano-additives can be used also to enhance the miscibility of PLA blends. The group of Amran et al. [252,253] compared the mechanical properties of a PLA/liquid natural rubber (LNR) blend (weight ratio of 90:10 wt/wt) reinforced with varying contents (0.25 wt.%, 0.50 wt.%, 0.75 wt.%, and 1.00 wt.%) of GNPs and functionalized GNPs. PLA/LNR exhibited a tensile strength 19.5 MPa, while the maximum tensile strength was demonstrated by the blend with 0.75 wt.% at 24.4 MPa. The same pattern was observed after the incorporation of the GNP-A and GNP-T fillers. In total, the 0.75 wt.% samples yielded the highest tensile strength at 45.47 MPa and 35.70 Mpa, respectively.

Wang et al. [254] exploited vinyl functionalized graphene (VGN) as an efficient compatibilizer for the preparation of mechanically strong biodegradable nanocomposites from PLA/PCL blends. After being reactively compatibilized using 0.5, 1.0, and 2.0 wt.% VGN, the phase size of co-continuous PLA/PCL blend was remarkably decreased, and the tensile strength was increased by 200%, 280%, and 253%, respectively. The strong interfacial interactions in reactively compatibilized PLA/PCL/VGN blend nanocomposites were evidenced by the linear rheological and atomic force microscopical modulus measuring results.

Different contents of functionalized graphene nanoplatelet (FGNP) were incorporated into a PLA/chitosan (PLA/CS) (75/25 wt/wt) blend and the effect of the filler on the mechanical performance, biodegradability, and electrical properties of the blend was investigated [255]. The nanocomposite filled with 3 phr FGNP indicated an improvement of approximately 142% and 261% in tensile strength and Young’s modulus, respectively, compared to the neat blend. Furthermore, with the addition of 3 phr FGNP to the PLA/CS blend, the thermal stability significantly improved. In addition, the study of biodegradability behavior demonstrated that the weight loss rate improved over time.

Wang et al. [256] prepared a PLA/poly(butylene succinate) (PBS) blend (weight ratio 70/30 wt/wt) and reduced graphene oxide (rGO) (in different contents of 0, 0.5, 1.0, 2.0, and 3.0 wt.%) as a filler via melt blending. The study showed an improvement of the mechanical properties, especially with the addition of 3.0 wt.% rGO. Elongation at break also increased to 9.2% by adding 3 wt.% rGO from 5.9% of the neat blend. Measurement of the tensile strength indicated an increase of around 65% compared to the neat PLA/PBS blend and the impact strength exhibited a value 2.9 times higher than the initial blend, owing to the good dispersion of the filler and the compatibilization of the components of the composites.

### 4.3. Applications of PLA/Nanocomposites with Natural Nano-Additives

#### 4.3.1. Food Packaging

Briefly, for food packaging applications, Montes et al. [257] extruded and successfully layered through thermo-compression PLA monolayer films containing Lgn-NPs and Umbelliferone (UMB). In vitro antioxidant studies using the DPPH method produced a radical scavenging activity (RSA) value of 80%, which corroborates with the results obtained by Yang et al. [258]. Moreover, the incorporation of Lgn-NPs decreased the transmittance in the visible region, reaching rather null values at wavelengths lower than 350 nm in the UV region. A food packaging material is expected to be transparent in visible light and opaque in the UV region, in order to protect the food from the oxidative deterioration, discoloration, and flavors losses causes by the UV radiation [259].

The UV, antioxidant, antibacterial, and compostability of PLA films containing 1 wt.% and 3 wt.% lignin nanoparticles (pristine (Lgn-NPs), chemically modified with citric acid (caLNP) and acetylated (aLNP)), were assessed by Cavallo et al. [130]. In general, and irrespective of whether Lgn-NPs were chemically modified or not, films containing Lgn-NPs restrained the growth of *Escherichia coli* and *Micrococcus luteus* when compared to neat PLA. After 17 days of disintegration studies in compost conditions, no evident amount of sample was observed, indicating the complete compostable nature of the composites. As expected, due to the presence of phenolic groups (-OH), ketones, and other chromophores, unmodified Lgn-NPs were capable of trapping DPPH radicals and exhibiting UV-blocking behavior, while for caLNP and aLNP, chemical modification involving their phenolic hydroxyl groups reduced both their antioxidant properties and UV protection. Finally, migration tests yielded significantly lower values than the migration limits allowed for food contact materials, thus rendering the nanocomposites suitable for the packaging sector.

PLA composite films were produced using unmodified soda micro- or nano-lignin as a green filler at four different contents, between 0.5 wt.% and 5 wt.% [260]. It was found that the tensile strength and the Young’s modulus were improved by the addition of lignin (L) and especially nanolignin (NL) (Figure 21a). This is due to the finer dispersion of NL in the PLA matrix, as verified by the TEM micrographs (Figure 21b). The UV-blocking and antioxidant properties of the composite films were also enhanced, especially at higher filler contents. As can be seen in Figure 21c, the residual DPPH content over time for different PLA–L (c) and PLA–NL (d) composites immersed in a DPPH/ethanol solution is lower in both L and NL composites, compared with neat PLA sample, which shows negligible activity. By contrast, the addition of either L or NL in the PLA matrix enhanced the antioxidant activity, as could be deduced from the higher reduction rate of DPPH• over time. The composites with higher L/NL content showed better antioxidant activity.

Similarly for food packaging purposes, Cerro et al. [129] developed PLA/Lgn-NPs nanocomposite films containing cinnamaldehyde (Ci) through a combination of melt extrusion and the supercritical impregnation process. The incorporation of Lgn-NPs and Ci affected the thermal, mechanical, and colorimetric properties of the developed films resulting in biodegradable plastic materials with a solid UV-light barrier performance compared to neat PLA films. Toxicity studies conducted upon rats presented normal blood parameters after a single dose of the nanocomposites. Disintegrability tests (Figure 22) under composting conditions verified the biodegradable character of developed materials. Moreover, the Ci active agent accelerated the disintegration rate, while the PLA films with Lgn-NPs presented a slightly reduced rate of disintegration. At Day 23, a disintegration degree higher than 90% was determined for all bio-nanocomposites.

The disintegration of PLA under composting conditions is one of the most appealing properties for food packaging purposes. Its degradation typically starts with the hydrolysis of the PLA chains induced by the diffusion of water into the composites. The influence over the degradation process strongly depends on the hydrophilicity/hydrophobicity and dispersion of the added NPs. Yang and his group [261] claimed in their work that the hydrophobic nature of Lgn-NPs (1 wt.%) delayed the disintegration process of the PLA matrix in a composting environment. Furthermore, when they added a 3 wt.% Lgn-NPs in the PLA matrix, nanoparticles aggregation and rougher film surface structures were detected, causing higher degradation.

Lgn-NPs have also been incorporated into PLA-based copolymers. In their study, Yang et al. [262] produced PLLA-PCL-lignin nanocomposites by a melt processing method, where Lgn-NPs grafted with PLLA and PCL copolymer served as interfacial compatibilizers.

Chihaoui et al. [263] examined the feasibility of incorporating lignocellulosic nanofibers (LCNFs) into a PLA/PEG blend (weight ratio of 80:20). Tensile tests and DMA indicated an enhancement in the tensile strength and Young’s modulus by about 250% and 1100%, respectively, at 8% LCNFs content while maintaining a high toughness (around 16 MJ/m^3^). The latter was attributed to the ability of the LCNFs to create a homogeneously distributed and entangled network within the PLA/PEG. The crystallization process of the blend was not affected by the addition of LCNFs, as no nucleation effects were observed. Finally, from the disintegration tests in composting conditions, both PEG and LCNFs were found to have a positive contribution to the disintegration of the PLA matrix, mostly owing to their hydrophilic nature favoring water diffusion inside the plasticized film (Figure 23).

Gulzar et al. [264] discovered electrospungelatin/chitosan solutions incorporated with tannic acid (TA) and chito-oligosaccharides (COS) at varying levels on PLA films. Bead-free and smooth nanofibers were formed at higher TA and COS levels. Both TA and COS provided antioxidative properties to the film, whereas a high amount of TA rendered higher antibacterial activity against both Gram-positive and Gram-negative bacteria on the surface of the films. Water vapor permeability (WVP), tensile strength, and elongation at break of GC-NF coated PLA films were found to be improved when compared to PLA film. Moreover, light transmission of GC-NF coated PLA films was reduced and the films were cloudy in appearance.

In another study, PLA was combined with poly(2-ethyl-2-oxazoline) (PEOx) and filled with nanocellulose extracted from corn cob (CCNC) [265]. Synthesized nanocomposites were prepared using the solvent casting method and they found that PEOx induces β-crystals formation while CCNC promotes α-crystals. In addition, the percentage of the improvement in the tensile strength and tensile modulus of PLA/PEOx/CCNC was higher than that of PLA/CCNC. This was attributed to PEOx, which facilitated the dispersion and distribution of CCNC in the PLA matrix.

Sangeetha et al. [266], investigated the effect of nanocellulose on properties of ethylene vinyl alcohol (EVOH)/ethylene vinyl acetate (EVA)-toughened PLA. Bio-nanocomposites were prepared via the melt mixing technique using a twin screw extruder followed by injection molding. The addition of nanofibrillated cellulose up to 2 wt.% retained the tensile strength and increased the tensile modulus of PLA/EVA/EVOH ternary blend systems, whereas further weight loadings decreased the tensile strength significantly. Furthermore, the addition of nanofibrillated cellulose increased the stiffness of the composite, raising the elongation at break up to 214% and the impact strength up to 89%. Additionally, they found that nanocellulose acted as a nucleating agent such that it initiated a crystallization phenomenon at a lower temperature.

Rasheed et al. [267] studied the effect of cellulose nanocrystals (CNC) from bamboo fiber on the properties of PLA/poly(butylene succinate) (PBS) composites fabricated by melt mixing. Results showed uniform distribution of CNC particles in the nanocomposites, improving their thermal stability, tensile strength, and tensile modulus up to 1 wt.%, making the composite films stiffer. The highest values of tensile modulus and tensile strength were obtained at 7600 MPa and 93 MPa for the composite with 1 wt.% of CNC. However, the elongation at break decreased insignificantly due to the reduction in flexibility of composites upon the addition of CNCs. In a thermal analysis, it was found that CNCs restricted the crystallization of PLA-PBS blends.

Rigotti et al. [150] investigated PLA nanocomposites containing various amounts (from 1 to 20 wt.%) of nanocellulose esterified with lauryl chains (LNC). Results showed that the low content of LNC (up to LNC content of 6.5 wt.%) in the PLA matrix was well dispersed and formed small, sub-micrometric clusters. In contrast, higher filler contents presented oval aggregates in the micrometric range. The addition of LNC did not affect the thermal properties (glass transition temperature and melting temperature) of the PLA matrix. Concurrently, as LNC content increased, both storage and elastic moduli presented a sharp decrease of up to 5 wt.% filler, and a lower reduction for LCN concentration of 10–20 wt.%. Nanocomposites with 3 and 5 wt.% filler exhibited the highest strain at break and a large amount of plastic deformation due to a strong interfacial adhesion between the filler particles and the PLA matrix. The addition of LNC fillers improved the gas barrier properties of the PLA film to a critical LNC concentration of 6.5 wt.%, where the gas permeability of the nanocomposite resulted in being 70% lower than that of the PLA matrix. The addition of higher LNC content increased the gas permeability of the nanocomposites due to the presence of large LNC aggregates.

Jin et al. [268] compounded PLA and NC at various ratios to prepare biocomposite films via the solution casting process. To enhance the compatibility of PLA/NCC blends, the NCC was intentionally subjected to graft modification by 3- aminopropyltriethoxysilane (KH-550). The increased silanized NCC (SNCC) was found to improve the light resistance, air permeability, thermal stability, and mechanical performance of the PLA- based composite films. Especially, compared to the pristine PLA sample, the obtained PLA-based composite films with 0.5 wt.% SNCC showed increases of 53.87% and 61.46% in the tensile strength and elongation at break, respectively. Additionally, PLA/SNCC composite films displayed a decrease of 87.9% in air permeability compared with neat PLA film.

Finally, in order to improve the compatibility of PLA/NCC blends, the NCC was deliberately subjected to graft modification by 3-aminopropyltriethoxysilane (KH-550). The obtained PLA nanocomposite with 0.5 wt.% NCC increased by 53.87%, and by 61.46% for the tensile strength and elongation at break, respectively, compared with PLA neat films. Moreover, PLA/NCC composites presented a decrease of 87.9% in air permeability in comparison to pure PLA film. In another case, nanocellulose was oxidated (TOBC) using TEMPO and Pickering emulsion in order to prevent the presence of aggregates of hydrophilic nanocellulose in the PLA matrix. It was found that the Pickering emulsion improved the dispersion of nanocellulose in the PLA matrix. TOBC improved the crystallization rate of PLA as the crystallinity of the composite materials containing 1.5% TOBC was 2.16 times higher than that of pure PLA. The addition of 1.5% TOBC in PLA also enhanced the mechanical strength and toughness of the materials. Specifically, the tensile strength, elongation at break, maximum bending strength, and elastic modulus of the PLA were raised by 9.2%, 202%, 45%, and 49%, respectively [269].

Another group reinforced PLA with oxidized nanocellulose by TEMPO-mediated and polyethylene glycol (PEG) to enhance the compatibility and strength of the PLA. The tensile strength of neat PLA decreased with the addition of oxidized nanocellulose due to incompatibility between the PLA matrix and oxidized nanocellulose. However, interestingly, the tensile modulus, which represents the resistance to deformation of material, of PLA/1PEG and PLA/1PEG/oxidized nanocellulose, were higher than PLA and PLA/oxidized nanocellulose, respectively, because the heterogeneous nucleation may occur so the PLA chains can form in an orderly fashion in the crystalline regions, which results in higher tensile modulus [270].

Sobhan et al. [271] prepared biochar nanoparticle BCNP/PLA nanocomposites by the solvent casting method. They studied the electrical conductivity of the BCNP/PLA nanocomposites and they found that increasing the BCNP content in nanocomposites from 50% to 85% increased the cyclic voltammetry (CV) and differential plus voltammetry (DPV) from 5 to 22 mA and 1.9 to 12 mA, respectively. Correspondingly, they proved that the addition of BCNP improved the thermal stability of nanocomposites, increasing the content of BCNP as 85% BCNP/PLA nanocomposite had the less thermal degradation at 400 °C. These results were also confirmed by Jasim et al. [272], who developed PLA/(0.5–10) wt.% BCNP nanocomposites using thymol as a plasticizer. Results showed that electrical conductivity increased with increased BCNP. Additionally, they noticed that BCNP increased the elongation and tear resistance but reduced the tensile strength and tensile modulus and hardness. Specifically, the Young modulus of PLA neat presented a value of 2.83 GPa, while the nanocomposites’ values ranged from 1.94 GPa with 0.5% biochar to 1.09 GPa with 10% biochar.

#### 4.3.2. Flame Retardants

Chollet and his group [273] examined for the first time the use of Lgn-NPs as a flame-retardant additive for PLA. Nanoparticles were grafted using diethyl chlorophosphate (LNP-diEtP) and diethyl (2-(triethoxysilyl)ethyl) phosphonate (LNP-SiP) to enhance their flame-retardant effect in the PLA matrix. In brief, they found that phosphorylated lignin nanoparticles restrained the PLA degradation during melt processing and the nanocomposites were shown to be relatively thermally stable. Even at a low concentration (5 wt.%), the grafting of phosphorus on the surface of Lgn-NPs enables a significant increase of ignition time as well as a reduction of peak of heat release rate (pHRR).

#### 4.3.3. Pesticides

In an attempt to reduce the groundwater contamination induced by traditional pesticides, leading to ecosystem destruction and food pollution, the group of Yu et al. [274] developed abamectin nanopesticide (Abam-PLA-Tannin-NS) and azoxystrobin nanopesticide (Azox-PLA-Tannin-NS) with strong adhesion to foliage via chemical modification. Hydrogen bonding was mainly responsible for the interaction between TA-coated NPs and the foliage. Abam-PLA-Tannin-NS and Azox-PLA-Tannin-NS exhibited better photostability and excellent continuous sustained release. The retention rates of Abam-PLA-Tannin-NS and Azox-PLA-Tannin-NS on the foliage surface were outstandingly improved by more than 50%, compared with untreated nanopesticides due to affinitive binding. Resultantly, the indoor toxicity of Abam-PLA-Tannin-NS and antifungal activity of Azox-PLA-Tannin-NS were enhanced.

#### 4.3.4. Implantable Medical Devices

Liao et al. [275] prepared filaments from PLA and acetylated tannin (AT) via a twin-screw extruder for 3D printing. The acetylation of tannin contributed to its well dispersion within the PLA matrix, guaranteeing the successful fabrication of PLA/AT composite filaments. Experimental results suggest that PLA can be compounded with up to 20 wt.% AT without any visible deterioration in the tensile property. The resulting composites acquired a better degradation rate in aquatic systems compared with neat PLA, especially in an alkaline medium. Moreover, the incorporation of various AT contents induced no noteworthy effects on the T_m_ and T_g_ of PLA because it has little influence on the intermolecular interactions or the chain flexibility of PLA polymer chains. A decrease of crystallinity was found in PLA/AT composites, leading to degradation rates in aquatic systems, particularly under alkaline environments, which would be beneficial in short-term applications such as implantable devices in the biomedical field.

The same group successfully prepared an entire biocomposite based on PLA and tannin acetate via in situ reactive extrusion using dicumyl peroxide (DCP) to enhance the interfacial adhesion [276]. The developed composites exhibited higher molecular weight as compared with reactive extruded PLA, together with a higher Young’s modulus and tensile strength. A rheological study showed that reactive extruded PLA/tannin acetate composites exhibited higher complex viscosity and modulus storage, indicating the strong interfacial adhesion of PLA matrix and tannin acetate. Moreover, T_g_, thermal stability, and crystallinity degree were all enhanced due to better interaction by radical initiated polymerization. Contact angle data revealed a more hydrophobic characteristic of the free radical extruded composites compared with pure PLA, which might be interesting for food packaging applications.

#### 4.3.5. Heavy Metals Removal

Kian et al. [277] also prepared PLA/PBS dual-layer membranes filled with a 0–3 wt.% cellulose nanowhisker (CNWs) with aim to remove metal ions from wastewater. The increase in CNWs from 0 to 3% loadings improved the membrane porosity (43–74%) but reduced pore size (2.45–0.54 μm). The thermal stability of neat membrane was enhanced by 1% CNW but decreased with loadings of 2 and 3% CNWs due to the flaming behavior of nanocellulose. Nanocomposites with 3% CNW displayed the highest tensile strength (23.5 MPa), Young’s modulus (0.75 GPa), and elongation at break (7.1%) as compared to other samples. Moreover, it exhibited the highest removal efficiency for both nickel and cobalt metal ions reaching 84% and 83%, respectively.

#### 4.3.6. Optical Applications

Geng et al. [278] synthesized PLA nanocomposites with poly(ethylene glycol) (PEG)-grafted cellulose nanofibers through a uniaxial drawing method to improve the dispersion ability of nanocellulose in the PLA matrix. With the incorporation of the PEG-grafted nanocellulose in PLA, the ultimate strength and toughness were enhanced by 39% and 70%, respectively, as compared to the nanocomposite containing unmodified cellulose nanofibers. Moreover, the aforementioned nanocomposites were highly transparent and possessed an anisotropic light scattering effect, revealing its significant potential for optical applications such as solar cells and displays.

#### 4.3.7. Outdoor Usages

Ghasemi et al. [279] synthesized PLA/NC composites through melt mixing using malleated PLA (PLA-g-MA) as a compatibilizer to facilitate the interaction between NC and the matrix. The morphology studies showed that a relatively good dispersion of NC was achieved within the PLA matrix in the presence of malleated PLA. However, mechanical studies pointed out that adding NC did not improve the impact strength of the nanocomposites compared to neat PLA, while using PLA-g-MA improved the nanocomposites’ impact strength significantly. In particular, with regard to samples containing 5% wt of PLA-g-MA, this increment would be 131% [280].

### 4.4. Nanoceramics

#### 4.4.1. Medical Applications

As previously stated, nanoceramics are extensively studied for medical applications because they are comparable to native bone mineral, exhibit high biocompatibility and have positive biological effects after implantation.

Specifically, Canales et al. [167] prepared PLA dense films with incorporated bioglass NPs via the melting process, at concentrations 5, 10, and 25 wt.%, obtaining nanocomposites with mainly a uniform distribution of nanometric particle agglomerates, although isolated particles could also be detected. This high level of particle dispersion demonstrates a good interaction between the polymer and nBGs, partially compensating the particle/particle interaction. The presence of nBGs increased the stiffness of the polymer matrix since the Young’s modulus increased with the increase of the nBGs content, and the 25 wt.% loading exhibiting the maximum value, about 52.6% higher than neat PLA. It was observed by thermal studies that although Tg did not show any significant change with the nanoparticle incorporation, the crystallization temperature decreased by 12.9 °C with the 25 wt.% nBGs loading, indicating that nanoparticles disrupted both the regularity of the PLA and therefore the crystallization process. The presence of nanoparticles also decreased the thermal stability of the PLA matrix, as nanocomposites presented up to about 20 °C lower degradation temperatures in a nitrogen atmosphere. The bioactive capacity of the PLA/nBGs composites increased with the increase of nanoparticles content, from 5 to 10 wt.%, inducing the formation of an apatite layer on the sample’s surface with a Ca/P ratio of 1.47, confirming the development of nBGs. However, the 25 wt.% loading PLA composite barely presented antibacterial behavior but exhibited higher viability on HeLa cells (cervical uterine adenocarcinoma cell line) compared to neat PLA.

Castro et al. [281] prepared fibrous scaffolds from PLA and ceramic nanobioglass (PLA/nBGs) at 5 and 10 wt.%. They observed a uniform distribution of the nBGs within the polymeric matrix and the increase in the concentration of the nBGs did not affect its internal sponge structure. Additionally, they noted that the incorporation of nBGs improved the thermal stability of the PLA due to the hydrogen bonds between the carbonyl group of the PLA and the hydroxyl groups of the nBGs and van der Waals interactions. Nanocomposites with 5 wt.% and 10 wt.% nBGs content exhibited a decrease in crystallinity by 71.8 and 71.1%, respectively. This decrease took place because nanocomposites do not have compact crystal lattices, with nucleation predominating on the material’s surface. The results from the biological analysis concluded that the incorporation of the nBGs did not affect the cell viability of HeLa cells. In the case of subdermal implantation analyses in biomodels, reabsorption was very slow, showing a significant presence in the material even three months after implantation, with an inflammatory response and an inflammatory capsule. Those changes helped the nanocomposites have higher reabsorption in the subdermal tissues than neat PLA, without affecting their biocompatibility. In addition, the nBGs induced the antimicrobial activity of Gram-positive and Gram-negative bacteria (e.g., Escherichia coli, Vibrio parahaemolyticus, and Bacillus cereus) at high concentrations of 20 *w*/*v*%, using the TTC method.

Esmaeilzadeh et al. [282] investigated the in vivo properties of PLLA/PCL/nBGs composites for a period six months for application to bioscrews. The in vivo results from the implants inserted on canine models indicated that the weight losses of the PLLA/PCL/n-BG composite and the PDLLA/PCL blend were approximately 60 and 70%, respectively. Moreover, the obtained histological images of the animal model after six months of implantation distinguished the formation of the new bone within the implanted area, while no osteitis, osteomyelitis, or structural abnormality were observed. They attributed the lower degradation rate of the nanocomposite compared to that of the neat blend implants to the presence of nBGs and their appropriate distribution throughout the PDLLA/PCL matrix. Further, it was proved that nBGs prevented the migration of the products resulting from PDLLA and PCL degradation. The acidity of the implants was caused by the acid release from products that were neutralized by the releasing of Ca-P ions of nBGs.

Macha et al. [283] developed PLA/3-aminopropyltriethoxysilane (APTES) nanosurface-modified nBGs composites via the solution casting method with different nBGs loadings (0.1, 0.5, and 1%). The modified nBGs provided better bonding between the amine groups of the APTES and carbonyl groups in the PLA matrix resulting in improved interfacial adhesion compared to composites with untreated particles. In addition, an improvement in elongation at break was detected, which indicated that nBGs modified by 1% APTES significantly influenced the percentage elongation of the PLA/BG nanocomposite at fracture by the effect of surface treating.

Hydroxyapatite (HAp) is the most emerging bioceramic, which is widely used in various biomedical applications, mainly in orthopedics and dentistry due to HAp having close similarities with the inorganic mineral component of bone and teeth.

Morsi et al. [284] filled the PLA matrix with iron-doped nHAp (FeHA NPs) in different mass fractions (PLA/FeHA NPs wt.%: 90/10, 80/20, and 70/30) via the solution casting technique for tissue engineering applications. The FT-IR results demonstrated that FeHA NPs were linked with PLA via hydrogen bond formation between OH of Ca(OH)_2_ and oxygen atoms in the ester group in PLA, which contributed to a better interface behavior. The inclusion of 10 wt.% FeHA NPs displayed the highest value of elastic modulus at 330 MPa, 28 MPa higher than the neat polymer.

Toong et al. [285] evaluated the nanocomposite formulations of nHAp and L-lactide functionalized nHAp and LA-nHAp with PLLA as potential thin-strut bioresorbable scaffolds. Functionalized nanofillers/PLLA exhibited improved nanofiller dispersion, and issues such as agglomeration and poor matrix/filler integration were mitigated. The uniform dispersion of nHAp improved the mechanical properties. Specifically, it was observed that the ultimate tensile strength of 10 wt.% nHAp/PLLA and 10 wt.% LA-nHAp/PLLA improved by 31.96 and 75.02 MPa, respectively, compared to neat PLLA, while 10 wt.% and 15 wt.% loadings of LA-nHAp in the matrix resulted in the highest values of elongation at break.

Michael et al. [169] modified nHAp using three different surface modifiers namely, 3-aminopropyl triethoxysilane (APTES), sodium n-dodecyl sulfate (SDS), and polyethylenimine (PEI) to improve the dispersion capacity of nHAp in the PLA matrix. The addition of 5 wt.% nHAp treated with APTES and SDS to the PLA matrix eliminated the presence of voids and agglomeration but increased the content of modified nHAp to 30 wt.%, which caused the nanofillers’ agglomeration. Similar behavior was observed in the modified nHAp with PEI. In addition, after the nHAp treatment with APTES and SDS, the tensile strength of PLA-5 wt.%-modified nHAp improved by 9 and 5 MPa, respectively, compared to the unmodified nHAp composite. On the other hand, treating nHAp with PEI caused a decrease of the tensile strength by 9 MPa compared to neat PLA. Similarly, PLA composites with APTES-nHAp and SDS-nHAp attained an improved tensile modulus of 8.5 and 2.4% in comparison with PLA neat, while PLA with PEI-nHAp recorded a decrease of 17.6%. This general improvement in tensile properties can be attributed to the improved interaction and homogenous dispersion of the modified nHAp in the PLA matrix. However, increasing modified nHAp loading to 30 wt.% caused the tensile strength to drop by 34% (APTES), 29.6% (SDS), and 40.5% (PEI), respectively, due to the presence of agglomeration.

Ziaee et al. [286] investigated the effect of HA and yttria-stabilized zirconia (YSZ), an inert and biocompatible ceramic with high bending and compressive strength and toughness, as bone tissue engineering scaffolds. An SEM analysis of the PLLA-HA-YSZ nanocomposites revealed a highly interconnected porous structure for all the scaffolds. The introduction of YSZ to the nanocomposite led to a significant increase in compressive strength, modulus, and densification strain due to toughening mechanisms and zirconia phase transformation from the tetragonal to the monoclinic. In addition, flexural strength and modulus showed an upward trend by the addition of YSZ particles to the scaffolds. It should be noted that PLA-20%HA-20%YSZ indicated the highest strength (0.47 MPa) and modulus (5.1 MPa) in both the compression and bending tests, although it did not demonstrate the proper strain compared to other scaffolds. Thus, PLA-15%HA-15%YSZ has been reported as the best candidate due to its appropriate strength and strain.

Ghassemi et al. [287] investigated the effect of adding different contents of nHAp on the mechanical properties of PLA/thermoplastic polyurethane (PLA/TPU) blends. They found that the addition of nHAp improved the interfacial properties, homogeneity, and compatibility of the blends. The 5wt.% nHAp-loaded nanocomposite indicated the highest tensile strength and modulus, i.e., an improvement of approximately 14 and 36%, respectively, compared to the PLA/TPU neat blend. On the other hand, the elongation at break of the blend diminished by the enhancement of the content of nHAp. They attribute this reduction to the fact that polymer chains were entrapped into HAp nanoparticles, which diminished the mobility of polymer chains because of the interactions. They also noted that another reason behind the decrease in the elongation-at-break was the introduction of stiff mineral nanoparticles, leading to stiffening of the blend nanocomposite.

Rahman et al. [288] examined the inclusion of chitosan and gelatin in a PLA/nanohydroxyapatite composite (GEL-CS-PLA/nHA), and discovered that the 5% nHAp-containing composite is the one most compatible with osteoconduction since it displayed a pore diameter of 125 μm, while the composites with 10%, 15%, and 20% nHAp revealed a smaller pore size in the range of 15–28 μm. Additionally, the addition of 10% nHAp in the GEL-CS-PLA blend showed the best compressive strength results as it was increased by 15.3% compared to the virgin blend. These results were due to the good dispersion of nHAp into the GEL-CHT-PLA blend and the compact morphology and less porous structure of nanocomposites. Regarding the thermal properties, the 10% nHAp-loading scaffold presented the greatest thermal stability (90–94% degradation at a temperature of 600 °C), while antibacterial and cytotoxicity test results revealed that the composite is resistant towards microbial attacks and has low sensitivity in cytotoxicity.

Yan et al. [289] prepared PLA/PBAT (poly(butylene adipat-co-terephthalte))/n-HA/MCC (microcrystalline cellulose) composites through physical blending. Good nanoparticle dispersion was observed when the nHAp content was lower than 6 wt.%, while obvious agglomeration of nHAp appeared by the addition of 10 wt.%. The tensile test demonstrated that by increasing the nHAp content, the tensile strength decreased gradually. However, the addition of 2 wt.% nHAp displayed the highest tensile strength compared with the other composites but was still lower than neat PLA/PBAT. This was because the addition of nHAp gradually increased the distance between the continuous phases of the material and weakened the interaction between the macromolecular chains. However, an improvement in thermal stability was observed along with the increase of the nHAp content from 2 wt.% to 10 wt.%.

The effect of integrating small amounts of nHAp (0.5, 1 and 3 wt.%) in a PLA/PCL (70:30 wt.%) blend was studied by Peponi et al. [290]. They found that the neat polymer blend reached elongations at break up to 200% due to the plasticizing effect of PCL on the PLA/PCL matrix, and these values were maintained for the nanocomposites up to 1 wt.% of nHAp, while the stretchability strongly decreased for the highest amount of nHAp (3 wt.%). Regarding the thermal properties of nanocomposites, they proved that the lowest amount of nHA (0.5 wt.%) did not produce significant changes on the thermal degradation process of nanocomposites. Higher amounts of nHA (1 wt.%) improved the thermal degradation of the PLA/PCL blend. Nevertheless, it should be highlighted that there was no degradation at temperatures below 200 °C which is the highest temperature used during processing.

Zhang et al. [168] reported uniform, high-quality oil-soluble HAp nanorods prepared by an innovative oleic acid (OA)-assisted mixed-solvent thermal method. The HAp nanorods were dispersed in PLA by a solution method in four concentrations (1, 3, 5, and 7 wt.%), followed by hot pressing to form nanocomposites films. As it turned out, the HA nanorods dispersed homogeneously as single particles without aggregation in the PLA matrix, which was attributable to the presence of an OA monolayer on the HA nanorods’ surface. Adding 3 wt.% HA nanorods elongation at break of the composites showed an increase of 4.55% compared to pure PLA, while 5 and 7 wt.% displayed a similar behavior to the neat polymer. This decrease was attributed to the higher crystallinity and crystal density. In addition, the 1 wt.% HA nanocomposite presented the highest tensile strength value, while the 3 wt.% displayed the highest Young’s modulus. However, the improvement of mechanical properties of PLA by the HA nanorods was not excellent, due to the low filler content and low length to diameter ratio o (L/D = 67.48).

Mondal et al. [291] used the 3D printing process to prepare scaffolds based on PLA and nHAp-modified PLA for medical applications. They concluded that the HAp nanoparticle accumulation on PLA scaffold surface enhanced its surface roughness and controlled the porosity. Specifically, the porosity of 3D-printed scaffolds with nHAp was decreased by 5–9%, compared to the neat PLA scaffold and SEM results (Figure 24), which confirmed the good distribution of nHAp on the PLA surface. The maximum compressive stress was detected in printed PLA-nHAp scaffolds at approximately 25 MPa, which was higher than the neat PLA scaffold. A thermogravimetric analysis revealed a significant thermal stability improvement after the addition of nHAp, considering that the maximum weigh loss was approximately 6.75% for PLA-nHAp and around 97% for PLA scaffolds. Additionally, it was found that cell attachment was highly influenced by the interaction of nHAp on the PLA scaffold surface, adsorbing proteins and facilitating cellular activity. On the contrary, the fabricated PLA scaffolds showed very poor cell attachment and proliferation on their surface.

In a similar case, Swetha et al. [292] fabricated 3D-printed PLA/Gel scaffolds and their surface was treated with Mg-nHAp in varying concentrations. Three-dimensional-PLA/Gel/Mg-nHAp scaffolds presented a uniform interconnected porous architecture. However, the porosity of the prepared scaffolds was decreased by the increasing of Mg-nHAp content (from 0.5 wt.% to 1.5 wt.%). The biological analysis showed that PLA/Gel/Mg-nHAp scaffolds exhibited improved protein adsorption in 24 h when compared with three-dimensional-PLA, which was attributable to their suitable porous architecture, as well as the presence of organic and inorganic components mimicking the biological bone matrix and apatite. It was also revealed that the PLA/Gel/Mg-nHAp three-dimensional-scaffolds were non-toxic, proving that the addition of Mg-nHAp had no influence on cell growth.

Wang et al. [55] discovered the capacity of PLA with nHAp (at a content of 0%, 10%, 20%, 30%, 40%, and 50%) to be 3D-printed in medical applications (Figure 25). The results showed that the PLA/nHAp composite had a stable structure, and the processing of this material was highly controllable. When the proportion of nHAp was greater than 50%, the composite material could not pass through the printing nozzle coherently and stably due to its high brittleness. When the nHAp ratio was less than or equal to 50%, the composite filament material could be printed by FDM. They also noticed a gradual change on the 3D-printed PLA/n-HA scaffolds’ surface from smooth to rough, as the nHAp ratio increased from 0% to 50% due to the appearance of a large number of convex and concave structures on the surface, which increased the surface area of the scaffold. After applying the 3D printing process to PLA/nHAp composites, the printed samples with 0% to 30% nHAp could still maintain the integrity of the structure after the pressure test, which proved that these samples maintained their elasticity. However, when the n-HA ratio reached more than 40%, the printed sample became brittle and could not maintain a complete its shape after compression. Nevertheless, the mechanical properties of all the PLA-printed scaffolds with the integration of nHAp were much higher than that of the pure HA porous ceramic and natural human cancellous bone. The biocompatibility and osteogenic induction properties were proven to be better than that of the pure PLA scaffold.

#### 4.4.2. Agricultural

Hajibeygi and Shafiei-Navid [293], taking advantage of nHAp’s surface functional groups such as P-OH and Ca-OH, used a new synthesized organic additive containing a-amino phosphonate and a naphthalene ring (PDA) as an organic modifier of HA NPs in order to increase the compatibility between PLA and nHAp. The presence of PDA led to improved dispersion of nHAp in the PLA matrix, while with the incorporation of only 2 wt.% nHAp and 6 wt.% PDA, the nanocomposite’s tensile strength was demonstrated to be 51 MPa higher than neat PLA. The initial decomposition temperature and char residue of PLA containing 6 wt.% PDA along with 2 wt.% nHAp were increased by 20 °C and 12%, respectively.

### 4.5. PLA/Nanoclays

#### 4.5.1. Food Packaging

Ramesh et al. [186] prepared PLA nanocomposites with 30% treated aloe vera fibers and 0, 1, 2, and 3 wt.% nanoclay fillers (PLA-TAF-MMT). They noted that the addition of 1 wt.% MMT in the PLA matrix acted as a fine bonding agent between the fibers and the matrix by the bonding of nano-matrix. On the other hand, the increased content of MMT clay (2–3% wt) showed an uneven surface, small voids, and agglomerations. The morphology of nanocomposites effected their mechanical properties. Specifically, the addition of 1 wt.% MMT in PLA improved the tensile and flexural modules by 18.84% and 60.95%, respectively, more than virgin-PLA. They attributed the increase in mechanical properties to the fact that the incorporation of MMT clay in PLA reduced micro-voids, presented uniform dispersion, and improved interaction, compatibility, and adhesion between the nano-matrix. On the contrary, the high amount of nanoclays (2–3% wt.) diminished the mechanical properties of PLA nanocomposites. The decomposition temperature increased by increasing the amount of MMT in the PLA matrix; this enhancement to the MMT clay acted as a barrier to the oxygen, constrained the mobility to the chain, and slowed down the decomposition process.

Similar results were presented by Othman et al. [180], who synthesized PLA nanocomposites incorporating different types (montmorillonite (MMT) and halloysite) and concentrations (0–9 wt.%) of nanoclays. PLA with 3 wt.% concentration of nanoclays resulted in the optimum mechanical and oxygen barrier properties due to the strong interaction between nanoclays and the PLA matrix. The incorporation of 3 wt.% nanoclays MMT and Halloysite in the PLA matrix raised the Young Modulus values to around 77% for PLA/MMT (2554 MPa) and 50% for PLA/Halloysite (2158 MPa) in comparison to the Young Modulus of neat PLA (1439 MPa). Nevertheless, these properties were decreased as more nanoclays (≥5 wt.%) were added into the films due to the aggregation of nanoclays. PLA/MMT nanocomposites showed better properties than PLA with halloysite nanoclays due to the nanoclay structure in nature. The addition of 3 wt.% nanoclays into virtually transparent PLA film had only minor effects on the transparency of the material as the reduction in light transmittance was only around 10%.

Alikarami et al. [26] researched the gas permeability of PLA/Cloisite-30B nanoclays composites for food packaging applications. They found that the addition of Cloisite-30B reduced the oxygen permeability by 55% compared to the pristine blend due to the tortuosity effect of nanosheets that were appropriately dispersed in the matrix. These results were confirmed by Salehi et al. [294], who synthesized poly(propylene) (PP)/PLA/nanoclay (Cloisite-30B) nanocomposite films and investigated their barrier properties against CO_2_, O_2_, and N_2_.

#### 4.5.2. Engineering Applications

Montava-Jorda et al. [192] prepared PLA composites with different halloysite nanotube (HNTs) loadings (3, 6, and 9 wt.%) with the presence of PVA compatibilizer for further uses as carriers for active compounds in medicine, packaging, and other sectors. The obtained results indicated a slight decrease in elongation at break as well as in tensile and flexural strength, with both properties being related to material cohesion; however, the reduction in mechanical properties was less than 7%. On the contrary, the stiffness increased with the HNTs content. Additionally, they noted that HNTs did not affect the glass transition temperature with invariable values of about 64 °C or the melt peak temperature. On the contrary, nanoclays moved the cold crystallization process towards lower values, from 112.4 °C for neat PLA down to 105.4 °C for the composite containing 9 wt.% HNTs.

Alakrach et al. [295] illustrated a similar trend with the addition of natural HNTs, proving that a small quantity of HNTs operated as plasticizer of PLA, which dispersed the impact energy during the break. Specifically, it was found that the incorporation of 4% wt HNTs increased the tensile strength by 43% and the elongation up to 4 wt.% compared to the neat PLA. However, Eryildiz et al. [295,296] proved that a lower amount than 4 wt.% HNTs in PLA was also able to increase the mechanical properties. Specifically, 0.5 wt.% HNTs in PLA raised the tensile strength about 50% and the elongation about 70%; as well, the addition of 2 wt.% modified HNTs improved the tensile strength to 15.3% and the elongation at break to 18.6%, greater than that of neat PLA. Moreover, Nizar et al. [297] researched the thermal properties of PLA/HNTs composites and they showed that the thermal stability of nanocomposites was reduced with the increasing addition of HNTs (5–8% wt) due to the presence of HNTs agglomerates in the PLA matrix. The uneven uniformity dispersion of nanoclays increased the thermal degradation of nanocomposites as the space between PLA and HNTs allowed the volatile pyrolyzed products of PLA to accumulate, thereby accelerating their degradation. Another research group melt mixed PLA with poly(ether-ester) elastomer (TPEE) and three loadings of BTN (1, 3, and 5% wt) bentonite (BTN) on a twin screw extruder, which was then injection-molded to improve the mechanical and thermal properties of PLA [298]. The Young’s modulus, impact strength, tensile strength, and thermal stability of all the prepared blends were enhanced, while the elongation at break deteriorated. Among the three PLA nanocomposites, the one with 1% wt BTN formed exfoliated structures, and therefore presented the highest elongation at break, tensile strength, and impact strength compared to the other synthesized nanocomposites.

## 5. Conclusions

The current review focuses specifically on the latest development of PLA-based nanocomposites. The demand for eco-friendly and sustainable products, exploitation of biobased feedstock, consideration of recycling routes, rising of oil price, and restrictions for the use of polymers with a high “carbon footprint” has paved the way for the swift expansion of biobased polymers. In this respect, PLA has been the most popular and promising of these biobased polymers due to its attractive renewability, biodegradability, and relatively low cost. Moreover, very recent studies underlined the huge potential of PLA in the case of higher added value for engineering applications, i.e., in food packaging, electronic and electrical devices, transportation, and mechanical and medical devices. Nevertheless, the PLA matrix must be improved in terms of thermal resistance, heat distortion temperature, and rate of crystallization while exhibiting some other specific properties (flame-retardancy, antistatic to conductive electrical characteristics, anti-UV, mechanical robustness antibacterial or barrier properties, etc.) to meet with end-user high demands. As presented in the current review paper, numerous nanofillers have been investigated towards that goal in order to prepare high-performance PLA-based nanocomposites using a variety of preparation methods.

Natural nanocompounds such as lignin and tannin NPs are especially suitable for active packaging applications since they accelerate the disintegration of PLA films and provide antioxidant and antimicrobial properties to prepared films. Incorporation of nano MgO imparts oxygen scavenger properties, limits oxidation, and enhances the shelf-life of foods, while ZnO NPs increase the UV shielding of food packaging films. Fabrication of PLA/TiO_2_ nanocomposites can be successfully employed for the photocatalytic removal of organic compounds from water streams. On the other hand, the addition of carbon-based fillers, such as CNTs and graphene, could boost the mechanical performance and electric properties of PLA-based materials in sensors and EMI shielding. Moreover, the capability of PLA-fullerenes nanocomposites to form supramolecular complexes with different types of molecules can be exploited in the development of organic photovoltaic cells.

Although a significant amount of research has been conducted on PLA nanocomposites, much work in certain fields is still required. The dispersion within the PLA matrix and the application of totally ‘’green’’ pathways and techniques pose a great challenge to academic societies and industries. Taking into consideration the above, it can be concluded that, undoubtedly, PLA-nanocomposites are currently under extensive investigation and their commercial utilization is rapidly catching up. Lastly, in-depth knowledge of these PLA nanocomposites will result in a new dimension in sustainable material development for numerous applications.

## Figures and Tables

**Figure 1 polymers-15-01196-f001:**
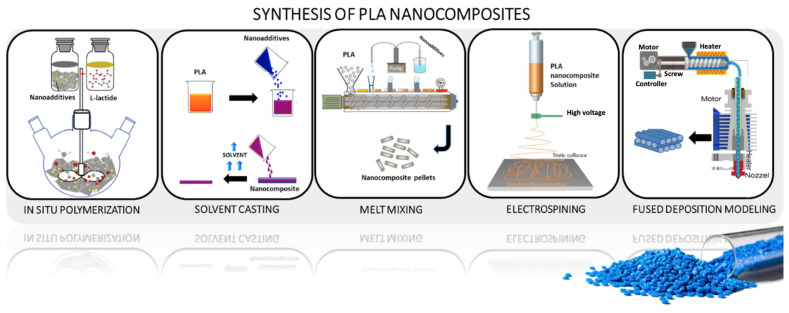
Synthetic routes employed for the preparation of PLA nanocomposites.

**Figure 2 polymers-15-01196-f002:**
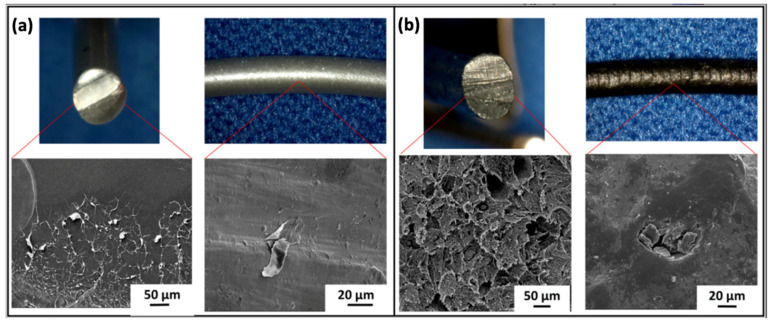
Top and cross-sectional microstructure of (**a**) neat PLA and (**b**) PLA-graphene filaments [57].

**Figure 3 polymers-15-01196-f003:**
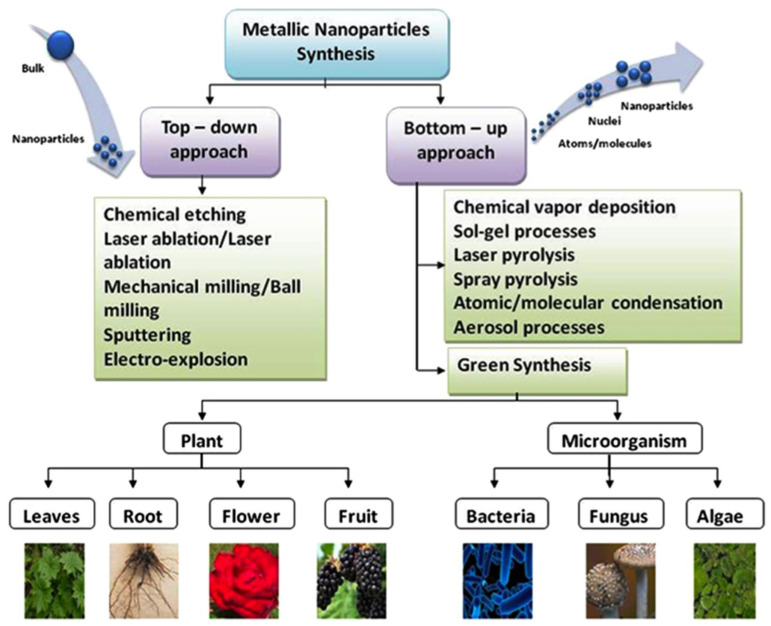
Top-down and bottom-up approaches for the fabrication of metal-based nanoparticles [62].

**Figure 5 polymers-15-01196-f005:**
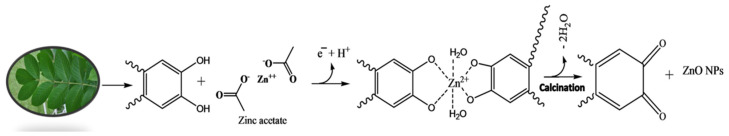
Biosynthesis of ZnO NPs using L.specioca leaves [72].

**Figure 6 polymers-15-01196-f006:**
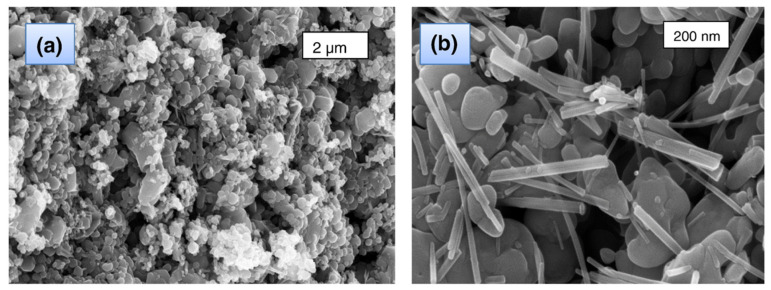
SEM analysis of ZnO-NPs: (**a**) 200 °C (spherical shape) and (**b**) 800 °C (nanorods) [72].

**Figure 7 polymers-15-01196-f007:**
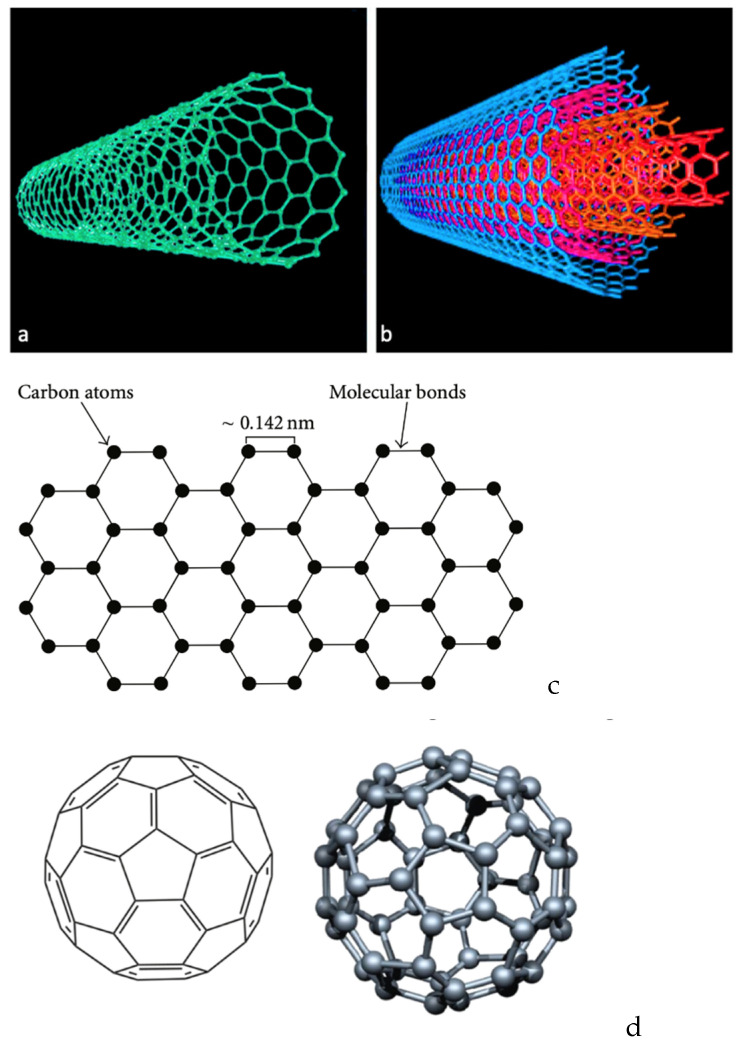
Carbon-based nano-additives. Three-dimensional presentation of (**a**) SWCNTs and (**b**) MWCNTs [110]; (**c**) schematic representation of a graphene sheet [111] and (**d**) 2D and 3D illustration of the fullerene C60 structure [112].

**Figure 8 polymers-15-01196-f008:**
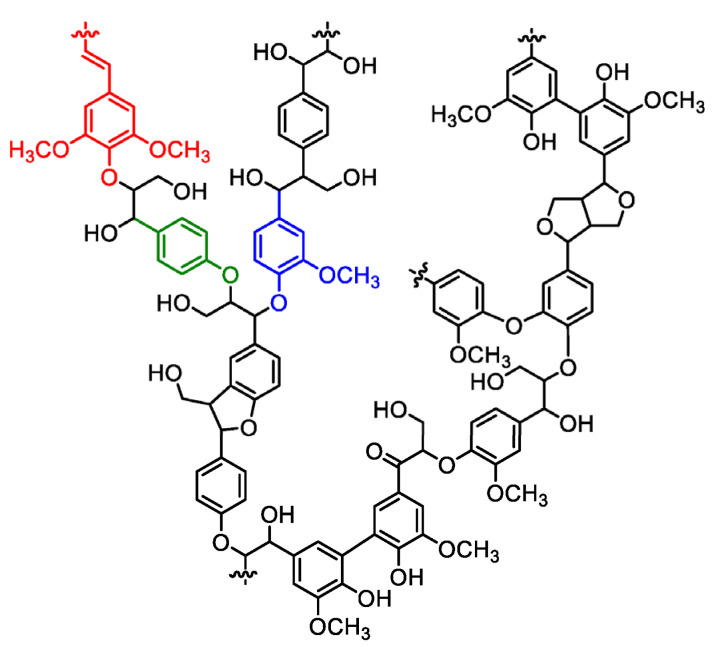
Chemical structure of lignin (Major monolignol units are colored: sinapyl alcohol—red, guaiacyl alcohol—blue, and p-coumaryl alcohol—green) [135].

**Figure 9 polymers-15-01196-f009:**
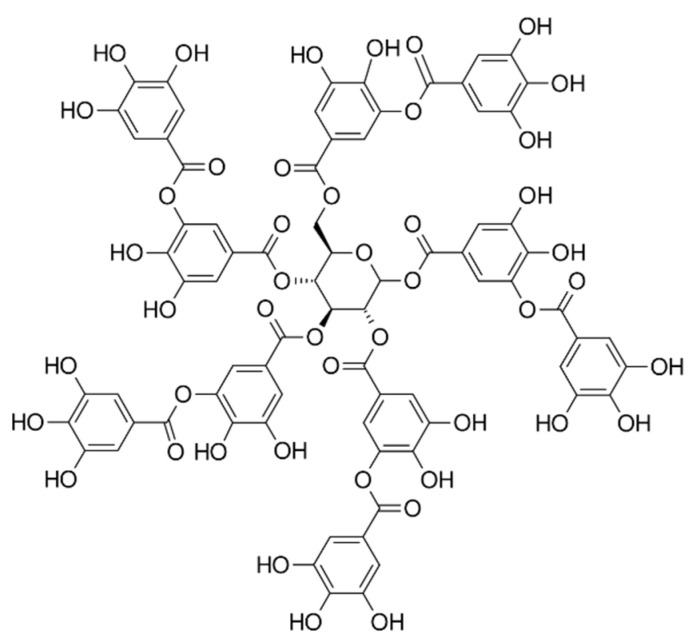
Chemical structure of tannin [137].

**Figure 10 polymers-15-01196-f010:**
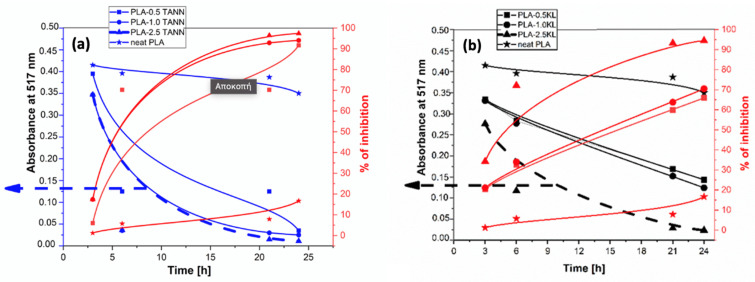
Reaction kinetics and % of inhibition of the PLA-based (**a**) TANN and (**b**) KL composites film evaluated with DPPH radical scavenging in the methanol solution indicated after 24 h [144].

**Figure 11 polymers-15-01196-f011:**
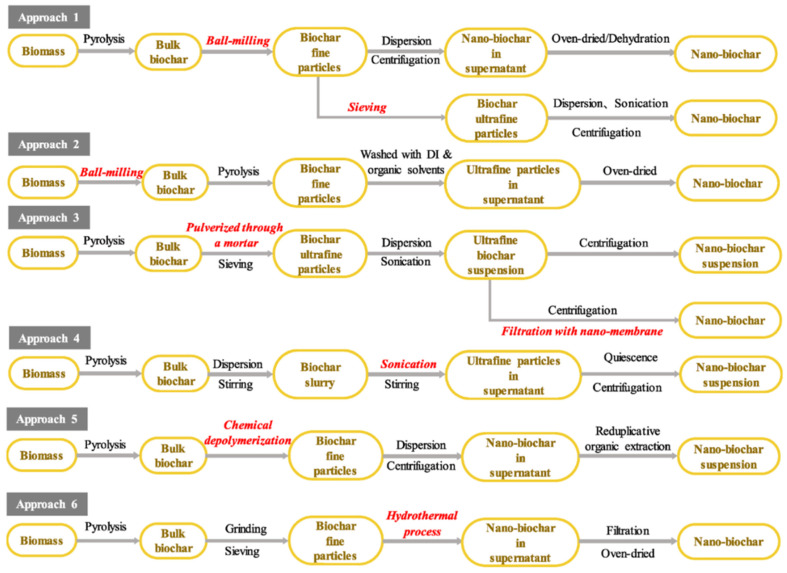
Commonly employed preparation routes for nano-biochar from biomass feedstock [159].

**Figure 12 polymers-15-01196-f012:**
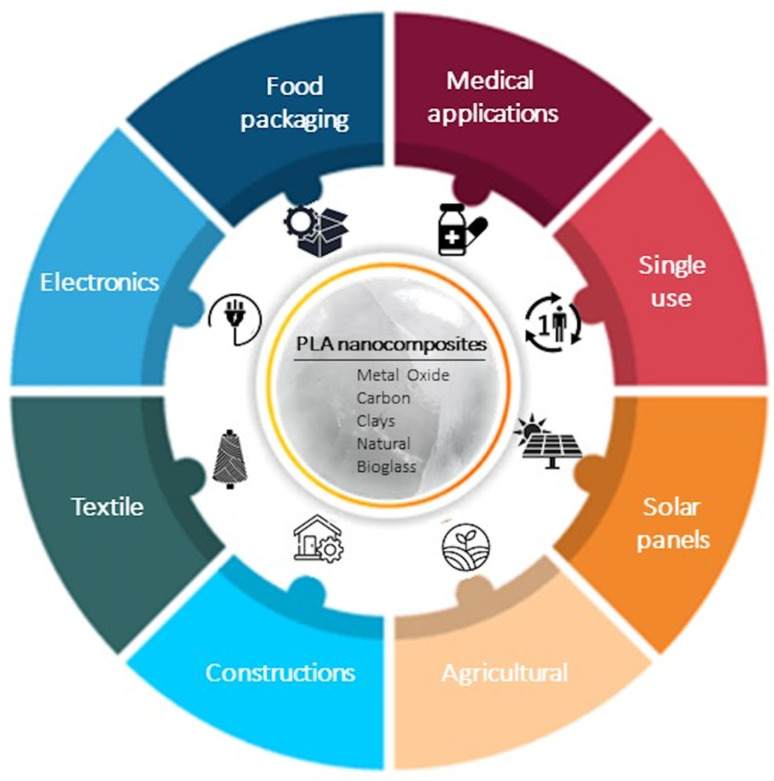
Schematic overview of the numerous applications involving PLA nanocomposites.

**Figure 13 polymers-15-01196-f013:**
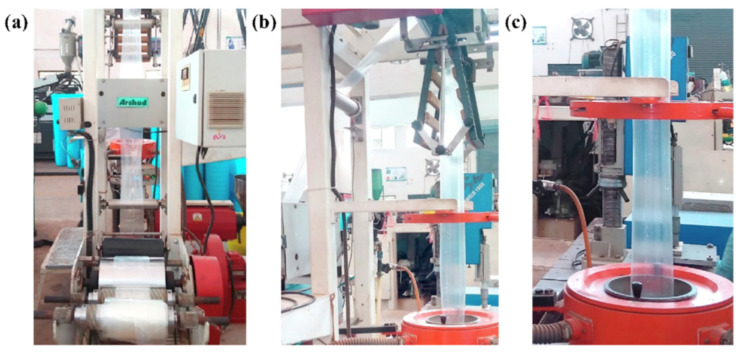
Industrial scale setup of the film blowing process: (**a**) Calendaring of the neat PLA film and (**b**,**c**) the blown film of the 1% PLA/MgO nanocomposite [194].

**Figure 14 polymers-15-01196-f014:**
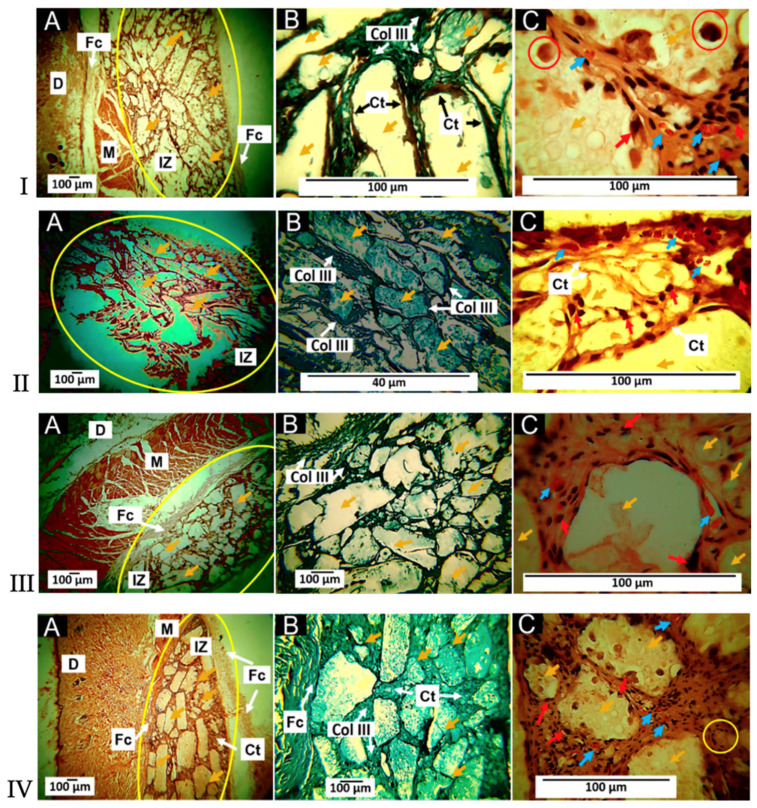
Subdermal implants of F1, F2, F3, and F4 at 60 days: (**A**) Formulations from the 4 X HE technique, (**B**) formulations from the 10 X GT technique, and (**C**) formulations from the 100 X HE technique. Yellow oval: Implantation zone. D: Dermis. M: Muscle. Fc: Fibrous capsule. IZ: Implantation zone. Col III: type III collagen. Yellow arrows: Fragments of materials. Red arrows: Inflammatory cells. Blue arrows: Blood vessels. HE: Hematoxylin and Eosin technique. GT: Gomori technique [199].

**Figure 15 polymers-15-01196-f015:**
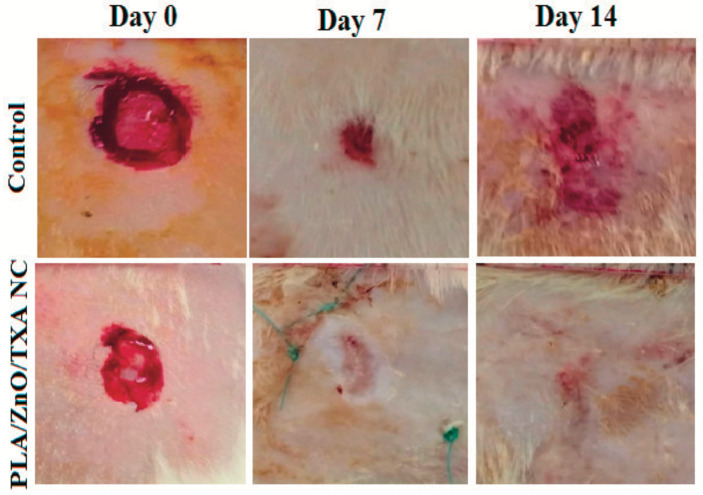
Images of skin wound healing process in mice after 0, 7, and 14 days of treatment with the PLA/ZnO/TXA nanocomposites [203].

**Figure 16 polymers-15-01196-f016:**
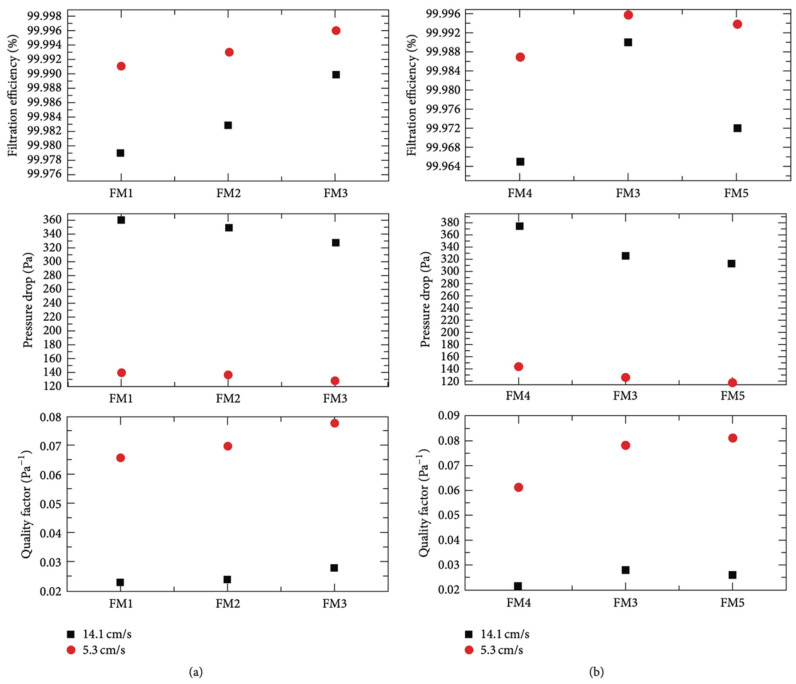
Filtration performance of FM1 (containing 0 wt.% TiO_2_ prepared at a relative humidity of 45%), FM2 (containing 1 wt.% TiO_2_ prepared at a relative humidity of 45%), FM3 (containing 1.75 wt.% TiO_2_ prepared at a relative humidity of 45%), FM4 (containing 1.75 wt.% TiO_2_ prepared at a relative humidity of 15%), and FM5 (containing 1.75 wt.% TiO_2_ prepared at a relative humidity of 60%) at various face velocities: (**a**) filtration efficiency, pressure drop, and quality factor of FM1, FM2, and FM3 at a face velocity of 5.3 cm/s and 14.1 cm/s, respectively and (**b**) filtration efficiency, pressure drop, and quality factor of FM3, FM4, and FM5 at a face velocity of 5.3 cm/s and 14.1 cm/s, respectively [213].

**Figure 17 polymers-15-01196-f017:**
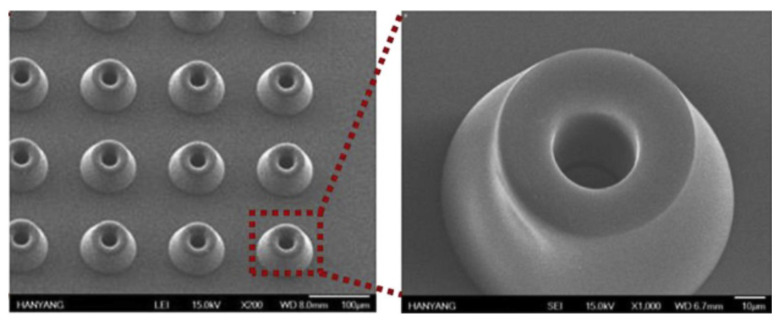
SEM micrographs of the micro needle surfaces of PLA/MWCNTs nanocomposites [232].

**Figure 18 polymers-15-01196-f018:**
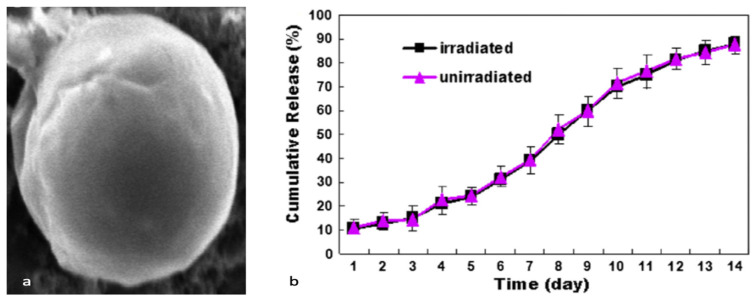
(**a**) SEM micrograph of the developed MTX microspheres and (**b**) in vitro release profiles of MTX from microspheres within 15 days of study [237].

**Figure 19 polymers-15-01196-f019:**
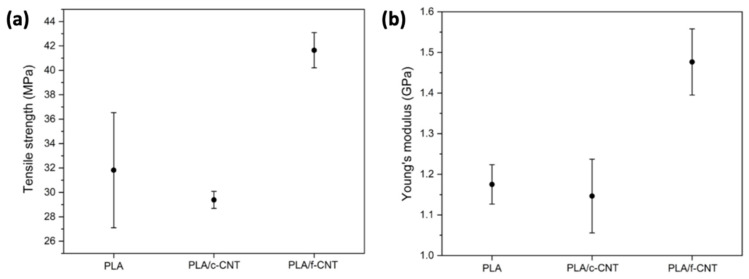
Effects of the c-CNT and f-CNT addition on the (**a**) tensile strength, and (**b**) Young’s modulus [238].

**Figure 20 polymers-15-01196-f020:**
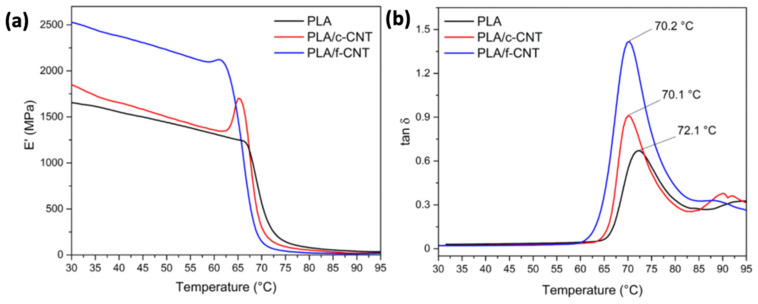
(**a**) Storage modulus (E′) and (**b**) Tan δ of pure PLA, PLA/c-CNT, and PLA/f-CNT obtained from DMA [238].

**Figure 21 polymers-15-01196-f021:**
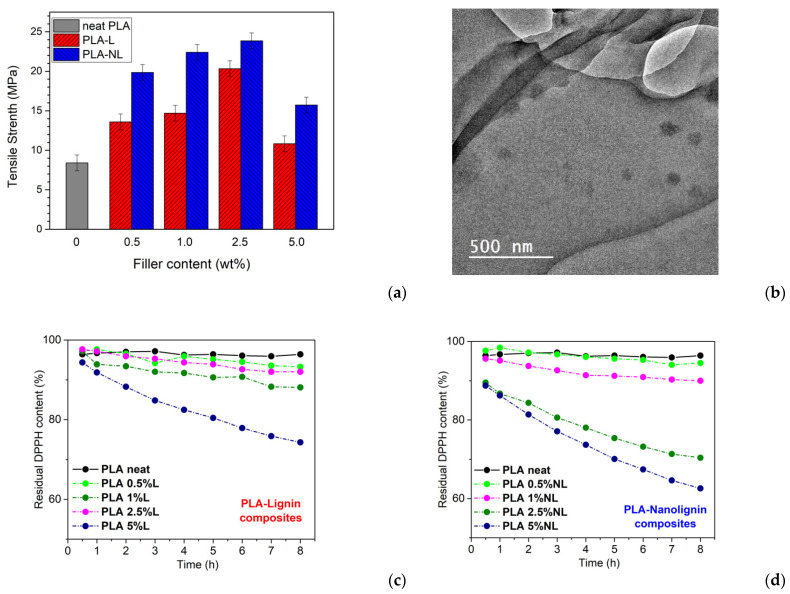
(**a**) Tensile strength variation pf PLA/lignin and nanolignin composites, (**b**) TEM micrographs of PLA/nanolignin containing 1 wt.% nanolignin and Reaction kinetics of the free radical DPPH during immersion of PLA–Lignin, (**c**) and PLA-Nanolignin (**d**) films in ethanol solution [260].

**Figure 22 polymers-15-01196-f022:**
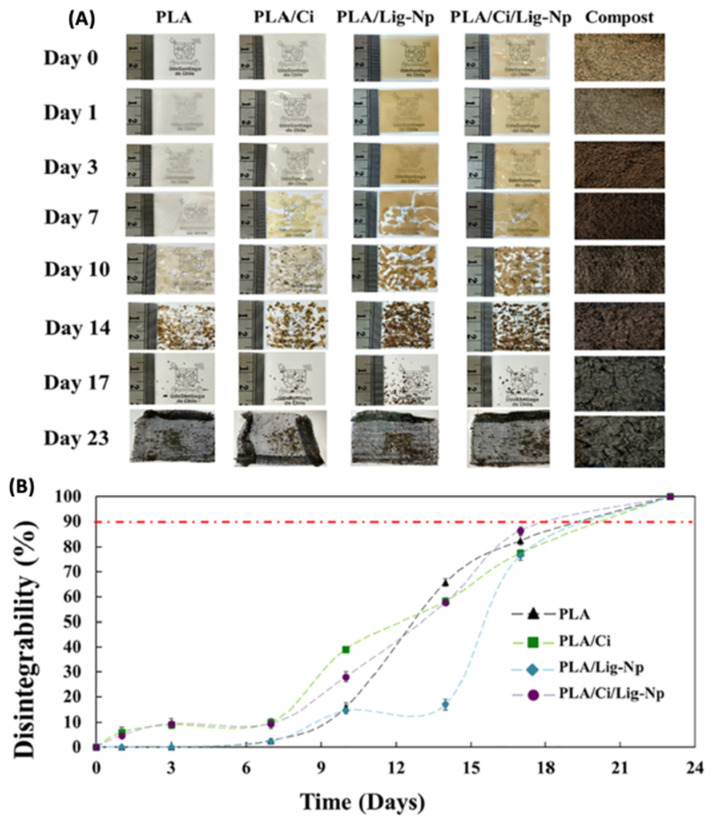
(**A**) Optical observation of prepared films before and after different incubation days under composting conditions and (**B**) % disintegrability degree under composting conditions [129].

**Figure 23 polymers-15-01196-f023:**
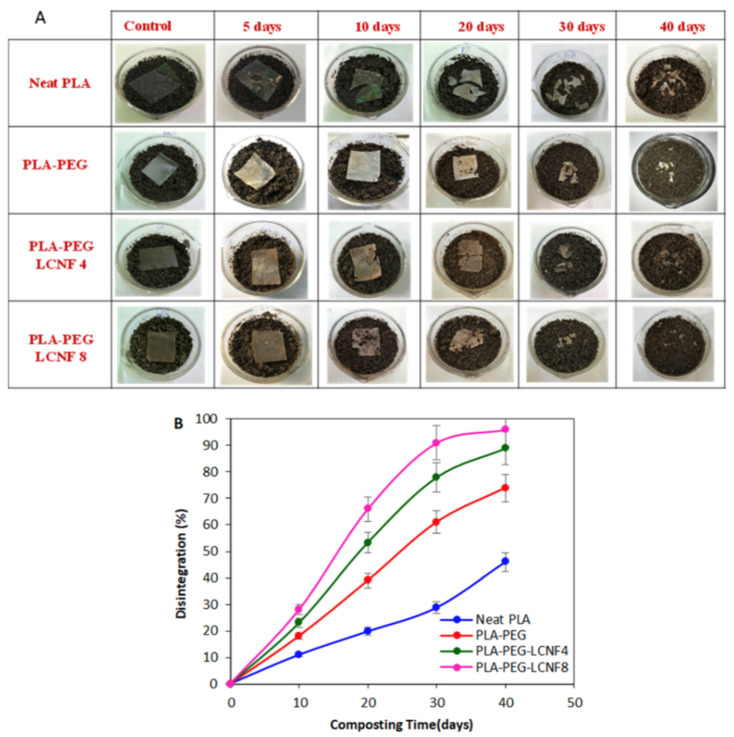
(**A**) Visual observation of PLA and PLA/PEG/LCNF nanocomposites after different days under composting environment and (**B**) disintegration degree under composting conditions as a function of time [263].

**Figure 24 polymers-15-01196-f024:**
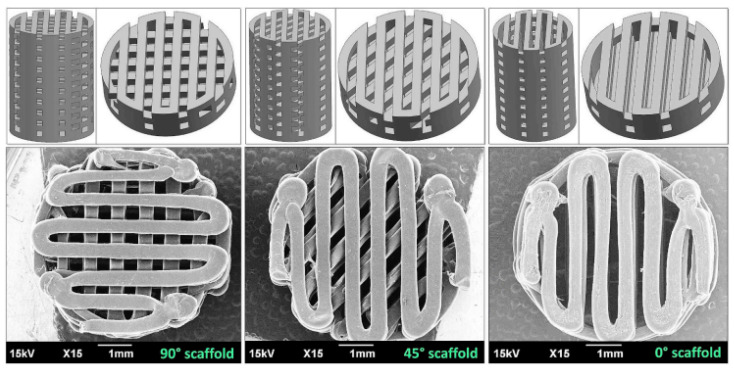
Three-dimensional-printed scaffolds destined for tissue engineering applications. The top view shows the schematic drawing from Solidworks 2016 (Dassault Systèmes, Vélizy-Villacoublay, France). The bottom view shows the original SEM image of the 3D-printed scaffold (MakerBot Z18 3D printer, New York City, New York, USA) [291].

**Figure 25 polymers-15-01196-f025:**
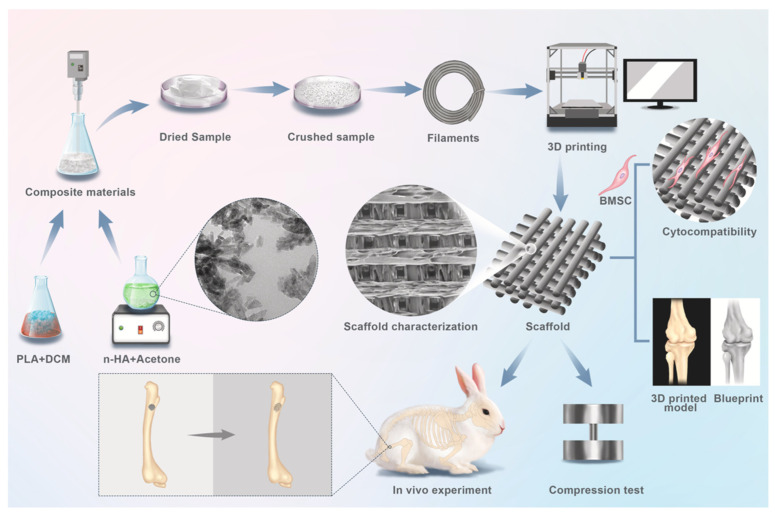
Schematic illustration of preparation and application of PLA/n-HA scaffolds for bone regeneration. The PLA/n-HA composite material was prepared by the wet mixing method, and the 3D-printed composite scaffold was made by PLA/n-HA filament. The material characterization and in vivo and in vitro experiments of the composite materials successively verified the sufficient mechanical strength and non-toxicity suitable for the promotion of the repair of bone defects [55].

**Table 1 polymers-15-01196-t001:** Varying sizes of spherical TiO_2_ NPs from different biological sources.

Biological Source	Size of NPs	References
Azadirachta indica, leaf extract	124 (average) nm	[66]
Aeromonas hydrophila, bacterium	28–54 nm	[68]
Annona squamosal, fruit peel extract	23 nm	[69]
Bacillus amyloliquefaciens, bacterium	15–86 nm	[70]
Euphorbia prostrata, leaf extract	83.22 nm	[71]

**Table 2 polymers-15-01196-t002:** Green synthetic sources of varying sizes of spherical ZnONPs.

Biological Source	Size	References
Jacarandamimosifolia, flow erextract	2–4 nm	[74]
Ruta graveolens, stem extract	~28 nm	[75]
Moringa oleifera, leaf extract	~6–10 nm	[76]
Polygala tenuifolia, root extract	33.03–73.48 nm	[77]
Sechium edule, leaf extract	36.2 nm (mean)	[78]

**Table 3 polymers-15-01196-t003:** PLA/nanoclays composites and their mechanical properties as found in the literature.

Sample	Content of Nanoclay	Tensile Strength	Young’s Modulus	Elongation at Break	Flexural Strength	Impact Strength	Bibliography
PLA/Halloysite	3% wt.	Increase of 14% compared to neat PLA	Increase of 50% compared to neat PLA	Increase of 3% compared to neat PLA	-	-	[180]
PLA/Kenaf fiber (30%)/MMT	1% wt.	Increase of 5.7% compared to PLA/Kenaf	Increase of 39.61% compared to neat PLA	-	Increase of 46.4% compared to PLA/Kenaf	Increase of 10.6% compared to PLA/Kenaf	[190]
PLA/Aloe vera fiber (30%)/MMT	1% wt.	Increase of 5.72% compared to PLA/Aloe vera	Increase of 18.84% compared to neat PLA	-	Increase of 6.08% compared to PLA/Aloe Vera	Increase of 10.43% compared to PLA/Aloe vera	[186]
PLA/Kenaf/Aloe vera/MMT	1% wt.	Increase of 23.2% compared to PLA/Kenaf Increase of 11.46% compared to PLA/Aloe vera	Tensile Modulus Increase of 24.61% compared to neat PLA	-	Increase of 56.43% compared to PLA/Kenaf Increase of 12.63% compared to PLA/Aloe vera	Increase of 57.5% compared to PLA/Kenaf Increase of 54.27% compared to PLA/Aloe vera	[187]
PLA/PCL/MMT	4% wt.	Increase of 15% compared to the blend	Increase of 26% compared to the blend	-	-	Decrease of 33% compared to the blend	[191]
PLA/Halloysite nanotubes (HNTs)	9% wt.	Decrease of 10.7% compared to neat PLA	Increase of 10.8% compared to neat PLA	Decrease of 46% compared to neat PLA	Decrease of 7.3% compared to neat PLA	Decrease of 51.4% compared to neat PLA	[192]

## Data Availability

The data presented in this study are available on request from the corresponding author.

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
