# Peer review of "Recent Advances in the Investigation of Poly(lactic acid) (PLA) Nanocomposites: Incorporation of Various Nanofillers and their Properties and Applications"

_polymers, 2023, doi:10.3390/polym15051196_

Round 1

Reviewer 1 Report

The submitted manuscript is very extensive. The collected information is presented in good form. The work has a logical arrangement. It allows you to take a broader look at the material which is PLA. The number of literature items is very large, it should be emphasized that it does not include information about all the research conducted. The literature is very current. The work, after development and additions, could constitute an independent study on the PLA. The drawings shown are well described. Tables are a good supplement to the main text.

Reviewer 2 Report

1. Page 5 line 170

"On the other hand, this method is unsuitable for PLA/graphene nanocomposites, since the pristine graphene has a tendency to agglomerate in polymer matrix." Please add what are the methods of creating PLA/ graphene nanocomposites into the paragraphy for comparisons.

2. Page 4 line 130

"The sonication method is often practiced aiding in the dispersion of nanofillers." Please add the reference to this.

3. Page 5 line 185

"The electrospinning machine set-up can be a single-nozzle or multi-nozzle type." Are there any references to support this?

The authors can consider to draw a scheme for these two types of set up.

4. Page 5 line 191

"Yet, this technique of fabricating the PLA fibers is mostly limited to fiber diameter in micro-scale, which is a constraint to PLA further applications." What are the constraint? The details of the size-dependent PLA fiber should discuss more specific.

5. Page 6 line 227

"Wang et al.,[47] researched the capacity of PLA with nHAp (at a content of 0%, 10%, 20%, 30%, 40%, and 50%) to be 3D printed for medical applications" The authors failed to mention what is the nHAp in the article?

6. Page 7 line 263

The "nanoparticles" has been assigned to NP so should be replaced as NP.

7. The authors mentioned a lot in how to prepare the metal oxide nanoparticles. However, the articles should focus on how these metal oxide nanoparticles affect or related to PLA and how do these nanoparticles change their properties in PLA composite. The author should add at least one paragraph to discuss.
